# Variance Reduced Policy Evaluation with Smooth Function Approximation

**Hoi-To Wai**
The Chinese University of Hong Kong
Shatin, Hong Kong
htwai@se.cuhk.edu.hk

**Mingyi Hong**
University of Minnesota
Minneapolis, MN, USA
mhong@umn.edu

**Zhuoran Yang**
Princeton University
Princeton, NJ, USA
zy6@princeton.edu

**Zhaoran Wang**
Northwestern University
Evanston, IL, USA
zhaoranwang@gmail.com

**Kexin Tang**
University of Minnesota
Minneapolis, MN, USA
tangk@umn.edu

## Abstract

Policy evaluation with smooth and nonlinear function approximation has shown great potential for reinforcement learning. Compared to linear function approximation, it allows for using a richer class of approximation functions such as the neural networks. Traditional algorithms are based on two timescales stochastic approximation whose convergence rate is often slow. This paper focuses on an offline setting where a trajectory of $m$ state-action pairs are observed. We formulate the policy evaluation problem as a non-convex primal-dual, finite-sum optimization problem, whose primal sub-problem is non-convex and dual sub-problem is strongly concave. We suggest a *single-timescale* primal-dual gradient algorithm with variance reduction, and show that it converges to an $\epsilon$-stationary point using $\mathcal{O}(m/\epsilon)$ calls (in expectation) to a gradient oracle.

## 1 Introduction

In reinforcement learning (RL) [39], policy evaluation aims to estimate the value function that corresponds to a given policy. It serves as a crucial step in policy optimization algorithms [19, 17, 34, 35] for solving RL tasks. Perhaps the most popular family of methods is temporal-difference (TD) [9], which estimates the value function by minimizing loss functions that are based on the Bellman equation. These methods can readily incorporate function approximations and have received huge empirical success, e.g., when the value functions are parametrized by deep neural networks [26, 36].

In contrast to the wide application of policy evaluation with nonlinear function approximation, most analytical results on policy evaluation focus on the linear setting [41, 40, 23, 14, 42, 45, 3, 37, 8]. However, when it comes to nonlinear function approximation, TD methods can be divergent [2, 43]. To remedy, Bhatnagar et al. [4] proposed an online algorithm for minimizing a generalized mean-squared projected Bellman error (MSPBE) with smooth and nonlinear value functions. Asymptotic convergence of this algorithm is established based on two-timescale stochastic approximation [5, 18] with diminishing step size. In a similar vein, Chung et al. [7] established the convergence of TD-learning with neural networks utilizing different step sizes for the top layer and the lower layers. However, non-asymptotic convergence results for nonlinear policy evaluation remains an open problem, illustrating a clear gap between theory and practice.

In this work, we make the first attempt to bridge this gap studying policy evaluation with smooth and nonlinear function approximation. We focus on the offline setting where we are provided with $m$

consecutive transitions from the policy to be evaluated, which is an important RL regime [20] and is closely related to the technique of experience replay [21]. Our contributions are two-fold:

- We recast the MSPBE minimization problem as a primal-dual optimization via the Fenchel's duality. Here, the objective function is a finite-sum, and is non-convex in the primal, strongly concave in the dual — constituting a *one-sided* non-convex primal-dual optimization problem.

- A variance reduced algorithm [cf. nPD-VR algorithm in Algorithm 1] is developed and applied to tackle the nonlinear policy evaluation problem. The algorithm performs primal-dual updates based on a single transition and has low computational complexity per-iteration. Unlike the existing algorithms, the proposed algorithm uses a fixed set of step sizes which is easier to tune and requires only a single for-loop to implement. We analyze the *non-asymptotic* performance of the algorithm and show that it converges to an $\epsilon$-stationary point of the MSPBE within $\mathcal{O}(m/\epsilon)$ calls to a gradient oracle, in expectation.

Note that the optimization problem arisen is strictly more challenging than plain non-convex minimization problems that are of recent interest, e.g., [30, 1, 15]. For instance, naive gradient-based updates for this problem might exhibit bizarre behaviors such as cycling [10]. To the best of our knowledge, the result in this paper constitutes the first convergence rate analysis for variance reduced policy evaluation with smooth and nonlinear function approximation.

**Related Work** Our work extends the research on policy evaluation with linear function approximation [41, 40, 23, 38, 14, 42, 8, 45, 3, 44, 6, 37, 13]; see [9] for a comprehensive review. Among these work, our work is closely related to [14, 44, 6], which study single- and multi-agent policy evaluation in the offline setting. Besides, they utilize the Fenchel's duality to obtain primal-dual optimization problems with a finite-sum structure, for which they provide variance-reduced optimization algorithms. Thanks to the linear function approximation, their objectives are strongly convex-concave, which enables linear rate of convergence. Furthermore, [4, 7] seem to be the only convergent policy evaluation results with nonlinear function approximation. Both of their algorithms utilize two-timescale step sizes, which may yield slow convergence. Moreover, their convergence results depends on two-timescale stochastic approximation [5, 18], which uses the trajectory of an ODE to approximate that of a stochastic process. When specialized to an offline setting similar to ours, [4] can be viewed as the primal-dual stochastic gradient algorithm for our problem.

From the optimization point of view, the non-convex primal-dual optimization (a.k.a non-convex min-max problem) that arise in the above non-linear policy evaluation setting is difficult to tackle. Although recent works have focused on the non-convex minimization problems [30, 1, 15], only a few have focused on the non-convex min-max problems. Recently, Daskalakis and Panageas [10], Daskalakis et al. [11] study the convergence of vanilla gradient descent/ascent (GDA), and the authors focused on bilinear problems (thus without the non-convex component). An optimistic mirror descent algorithm is proposed in [25], and its convergence to a saddle point is established under certain strong coherence assumption. In [28], algorithms for robust machine learning problems have been proposed, where the problem is linear in one side and non-convex in another side. In [29], a proximally guided stochastic mirror descent method (PG-SMD) is proposed, which updates the variables simultaneously, while adopting a double loop update rule in which the variables are updated in "stages". These algorithms yield the convergence rate in the order of $\mathcal{O}(1/\sqrt{K})$ and $\mathcal{O}(1/K^{1/4})$, respectively. Recently, an oracle based non-convex stochastic gradient descent for generative adversarial networks was proposed in [31, 32], where the algorithm solves the maximization subproblem up to some small error. It was shown in [24] that a deterministic gradient descent/ascent min-max algorithm has $\mathcal{O}(1/K)$ convergence rate. In [27], the $\mathcal{O}(1/K)$ convergence rate was proved for nonconvex-nonconcave min-max problems under Polyak-Łojasiewicz conditions.

**Organization** In §2 we describe the setup for the policy evaluation problem with smooth (possibly nonlinear) function approximation. In §3 we describe the variance reduced method for policy evaluation and present the results from a preliminary numerical experiment. In §4 we provide the main convergence result in this paper for the proposed variance reduced method, in which a few key lemmas and a proof outline will be presented.

## 2 Markov Decision Process and Nonlinear Function Approximation

Consider a Markov Decision Process (MDP) defined by $(\mathcal{S}, \mathcal{A}, \mathcal{P}, \mathcal{R}, \gamma)$. We have denoted $\mathcal{S}$ as the state space and $\mathcal{A}$ as the action space, notice that both $\mathcal{S}, \mathcal{A}$ can be infinite. Let $s \in \mathcal{S}, a \in \mathcal{A}$ be a state and an action, respectively. For each $a \in \mathcal{A}$, the operator $\mathcal{P}^a$ is a Markov kernel describing the state transition *upon* taking action $a$. For any measurable function $f$ on $\mathcal{S}$, we have

$$\left(\mathcal{P}^a f\right)(s) = \int_{\mathcal{S}} \mathcal{P}^a(s, s') f(s') \mu(ds'), \ \forall \, s \in \mathcal{S}. \tag{1}$$

Lastly, the reward function $\mathcal{R}(s, a)$ is the reward received after taking action $a$ in state $s$ and $\gamma \in (0, 1)$ is the discount factor.

A policy $\pi$ is defined through the conditional probability $\pi(a|s)$ of taking action $a$ given the current state $s$. Given a policy $\pi$, the expected instantaneous reward at state $s$ is defined as:

$$R^\pi(s) := \mathbb{E}_{a \sim \pi(\cdot|s)} \big[ \mathcal{R}(s, a) \big], \ \forall \, s \in \mathcal{S}. \tag{2}$$

In the policy evaluation problem, we are interested in the value function $V : \mathcal{S} \to \mathbb{R}$ that is defined as the discounted total reward over an infinite horizon with the initial state fixed at $s \in \mathcal{S}$:

$$V(s) := \mathbb{E}\Big[ \textstyle\sum_{t=0}^{\infty} \gamma^t \mathcal{R}(s_t, a_t) | s_0 = s, a_t \sim \pi(\cdot|s_t), \ s_{t+1} \sim \mathcal{P}^{a_t}(s_t, \cdot) \Big]. \tag{3}$$

Let $\mathcal{M}(\mathcal{S})$ be the manifold of value function given the state space $\mathcal{S}$, we define the Bellman operator $\mathcal{T}^\pi : \mathcal{M}(\mathcal{S}) \to \mathcal{M}(\mathcal{S})$ as:

$$\left(\mathcal{T}^\pi f\right)(s) := \mathbb{E}\Big[ \mathcal{R}(s, a) + \gamma f(s') | a \sim \pi(\cdot|s), \ s' \sim \mathcal{P}^a(s, \cdot) \Big], \ \forall \, s \in \mathcal{S}, \tag{4}$$

where $f$ is any measurable function defined on $\mathcal{S}$. Denote $V(s)$ (*resp.* $R^\pi(s)$) as the average reward for the policy when initialized at a state $s \in \mathcal{S}$. The Bellman equation [39] shows that the value function $V : \mathcal{S} \to \mathbb{R}$ satisfies

$$V(s) = R^\pi(s) + \gamma \big( \boldsymbol{P}^\pi V \big)(s) = \mathcal{T}^\pi V(s), \ \forall \, s \in \mathcal{S}, \tag{5}$$

where we have defined the operator $\boldsymbol{P}^\pi(\cdot, \cdot)$ as the expected Markov kernel of the policy $\pi$:

$$\boldsymbol{P}^\pi(s, s') := \int_{\mathcal{A}} \mathcal{P}^a(s, s') \pi(a|s') \mu(da), \ \ \forall \, s, s' \in \mathcal{S} \times \mathcal{S}. \tag{6}$$

In the above sense, the policy evaluation problem refers to solving for $V : \mathcal{S} \to \mathbb{R}$ which satisfies (5).

### 2.1 Nonlinear Function Approximation

Solving for the function $V : \mathcal{S} \to \mathbb{R}$ in (5) is a non-trivial task since the state space $\mathcal{S}$ is large (or even infinite) and the expected Markov kernel $\boldsymbol{P}^\pi(\cdot, \cdot)$ is unknown. To address the first issue, a common approach is to approximate $V(s)$ by a parametric family of functions.

This paper considers approximating $V : \mathcal{S} \to \mathbb{R}$ from the family of parametric and smooth functions given by $\mathcal{F} = \{V_{\boldsymbol{\theta}} : \boldsymbol{\theta} \in \Theta\}$, where $\boldsymbol{\theta}$ is a $d$-dimensional parameter vector and $\Theta$ is a compact, convex subset of $\mathbb{R}^d$. Note that $\mathcal{F}$ forms a differentiable manifold. For each $\boldsymbol{\theta}$, $V_{\boldsymbol{\theta}}$ is a map from $\mathcal{S}$ to $\mathbb{R}$ and the function is non-linear *w.r.t.* $\boldsymbol{\theta}$. As we consider the family of smooth functions, the gradient and Hessian of $V_{\boldsymbol{\theta}}(s)$ *w.r.t.* $\boldsymbol{\theta}$ exists and they are denoted as

$$g_{\boldsymbol{\theta}}(s) := \left(\nabla_{\boldsymbol{\theta}} V_{\boldsymbol{\theta}}\right)(s) \in \mathbb{R}^d, \ \ H_{\boldsymbol{\theta}}(s) := \left(\nabla_{\boldsymbol{\theta}}^2 V_{\boldsymbol{\theta}}\right)(s) \in \mathbb{R}^{d \times d}, \tag{7}$$

for each $s \in \mathcal{S}$ and $\boldsymbol{\theta} \in \Theta$. We define $\boldsymbol{G}_{\boldsymbol{\theta}} := \mathbb{E}_{s \sim p^\pi(\cdot)}[g_{\boldsymbol{\theta}}(s) g_{\boldsymbol{\theta}}^\top(s)] \in \mathbb{R}^{d \times d}$, where $p^\pi(\cdot)$ is the stationary distribution of the MDP under policy $\pi$. Throughout this paper, we assume that $\boldsymbol{G}_{\boldsymbol{\theta}}$ is a positive definite matrix for all $\boldsymbol{\theta} \in \Theta$.

To find the best parameter $\boldsymbol{\theta}^\star$ such that $V_{\boldsymbol{\theta}^\star} : \mathcal{S} \to \mathbb{R}$ is the closest approximation to a value function $V$ that satisfies (5), Bhatnagar et al. [4] proposed to minimize the mean squared projected bellman error (MSPBE) defined as follows:

$$J(\boldsymbol{\theta}) := \tfrac{1}{2} \big\| \Pi_{\boldsymbol{\theta}} \big( \mathcal{T}^\pi V_{\boldsymbol{\theta}} - V_{\boldsymbol{\theta}} \big) \big\|_{p^\pi(\cdot)}^2, \tag{8}$$

where the weighted norm $\|V\|_{p^\pi(\cdot)}^2 = \int_{\mathcal{S}} p^\pi(s)|V(s)|^2\mu(ds)$ is defined with the stationary distribution $p^\pi(s)$ and $\Pi_{\boldsymbol\theta}$ is a projection onto the space of nonlinear functions $\mathcal{F}$ w.r.t. the metric $\|\cdot\|_{p^\pi(\cdot)}$, i.e., for any $f : \mathcal{S} \to \mathbb{R}$, we have $\Pi_{\boldsymbol\theta} f = \arg\min_{V_{\boldsymbol\theta}\in\mathcal{F}} \|f - V_{\boldsymbol\theta}\|_{p^\pi(\cdot)}^2$. The following identities are shown in [4]:

$$
\begin{aligned}
J(\boldsymbol\theta) &= \frac{1}{2}\mathbb{E}_{s\sim p^\pi(\cdot)}\big[(\mathcal{T}^\pi V_{\boldsymbol\theta}(s) - V_{\boldsymbol\theta}(s))g_{\boldsymbol\theta}(s)^\top\big]\,\boldsymbol{G}_{\boldsymbol\theta}^{-1}\,\mathbb{E}_{s\sim p^\pi(\cdot)}\big[(\mathcal{T}^\pi V_{\boldsymbol\theta}(s) - V_{\boldsymbol\theta}(s))g_{\boldsymbol\theta}(s)\big]\\
&= \frac{1}{2}\Big\|\mathbb{E}_{s\sim p^\pi(\cdot)}\big[(\mathcal{T}^\pi V_{\boldsymbol\theta}(s) - V_{\boldsymbol\theta}(s))g_{\boldsymbol\theta}(s)\big]\Big\|_{\boldsymbol{G}_{\boldsymbol\theta}^{-1}}^2\\
&= \max_{\boldsymbol{w}\in\mathbb{R}^d}\Big(-\frac{1}{2}\mathbb{E}_{s\sim p^\pi(\cdot)}\big[(\boldsymbol{w}^\top g_{\boldsymbol\theta}(s))^2\big] + \big\langle \boldsymbol{w}, \mathbb{E}_{s\sim p^\pi(\cdot)}\big[(\mathcal{T}^\pi V_{\boldsymbol\theta}(s) - V_{\boldsymbol\theta}(s))g_{\boldsymbol\theta}(s)\big]\big\rangle\Big)
\end{aligned}
\tag{9}
$$

where the last equality is due to the Fenchel's duality. With the above equivalence, the MSPBE minimization problem can be reformulated as a primal-dual optimization problem:

$$
\min_{\boldsymbol\theta\in\Theta}\max_{\boldsymbol{w}\in\mathbb{R}^d} L(\boldsymbol\theta, \boldsymbol{w}), \quad \text{where}
\tag{10}
$$

$$
L(\boldsymbol\theta, \boldsymbol{w}) := \big\langle \boldsymbol{w}, \mathbb{E}_{s\sim p^\pi(\cdot)}\big[(\mathcal{T}^\pi V_{\boldsymbol\theta}(s) - V_{\boldsymbol\theta}(s))g_{\boldsymbol\theta}(s)\big]\big\rangle - \frac{1}{2}\mathbb{E}_{s\sim p^\pi(\cdot)}\big[(\boldsymbol{w}^\top g_{\boldsymbol\theta}(s))^2\big].
\tag{11}
$$

For convenience, we call $\boldsymbol\theta$ as the primal variable and $\boldsymbol{w}$ as the dual variable. For any fixed $\boldsymbol\theta \in \Theta$, the function $L(\boldsymbol\theta, \boldsymbol{w})$ is strongly concave in $\boldsymbol{w}$ since $\boldsymbol{G}_{\boldsymbol\theta}$ is positive definite. Moreover, the primal and dual gradients are given respectively by:

$$
\begin{aligned}
\nabla_{\boldsymbol\theta} L(\boldsymbol\theta, \boldsymbol{w}) &= \mathbb{E}_{s\sim p^\pi(\cdot)}\big[\big(\mathcal{T}^\pi V_{\boldsymbol\theta}(s) - V_{\boldsymbol\theta}(s) - g_{\boldsymbol\theta}^\top(s)\boldsymbol{w}\big)H_{\boldsymbol\theta}(s)\boldsymbol{w}\big]\\
&\quad + \mathbb{E}_{s\sim p^\pi(\cdot)}\big[(g_{\boldsymbol\theta}^\top(s)\boldsymbol{w})\big(\gamma\mathbb{E}_{s'\sim p^\pi(\cdot|s)}[g_{\boldsymbol\theta}(s')] - g_{\boldsymbol\theta}(s)\big)\big],
\end{aligned}
\tag{12}
$$

$$
\nabla_{\boldsymbol{w}} L(\boldsymbol\theta, \boldsymbol{w}) = \mathbb{E}_{s\sim p^\pi(\cdot)}\big[(\mathcal{T}^\pi V_{\boldsymbol\theta}(s) - V_{\boldsymbol\theta}(s))g_{\boldsymbol\theta}(s)\big] - \mathbb{E}_{s\sim p^\pi(\cdot)}\big[g_{\boldsymbol\theta}(s)g_{\boldsymbol\theta}^\top(s)\boldsymbol{w}\big].
$$

The above follows from the gradient of the temporal difference error:

$$
\nabla_{\boldsymbol\theta}\big(\mathcal{T}^\pi V_{\boldsymbol\theta}(s) - V_{\boldsymbol\theta}(s)\big) = \gamma\mathbb{E}[g_{\boldsymbol\theta}(s')|s'\sim p(\cdot|s,a),\ a\sim p(\cdot|s)] - g_{\boldsymbol\theta}(s)
\tag{13}
$$

and we have denoted $\mathbb{E}_{s'\sim p^\pi(\cdot|s)}[g_{\boldsymbol\theta}(s')]$ as the expectation considered in the above. The primal dual projected gradient algorithm proceeds as

$$
\begin{aligned}
\boldsymbol\theta^{(k+1)} &= \mathcal{P}_\Theta\big\{\boldsymbol\theta^{(k)} - \alpha_{k+1}\nabla_{\boldsymbol\theta}L(\boldsymbol\theta^{(k)}, \boldsymbol{w}^{(k)})\big\}\\
\boldsymbol{w}^{(k+1)} &= \boldsymbol{w}^{(k)} + \beta_{k+1}\nabla_{\boldsymbol{w}}L(\boldsymbol\theta^{(k)}, \boldsymbol{w}^{(k)}),
\end{aligned}
\tag{14}
$$

where $\mathcal{P}_\Theta$ denotes the Euclidean projection onto the set $\Theta$. Applying the primal dual gradient algorithm (14) is difficult as evaluating the gradients $\nabla_{\boldsymbol\theta}L(\boldsymbol\theta^{(k)}, \boldsymbol{w}^{(k)}), \nabla_{\boldsymbol{w}}L(\boldsymbol\theta^{(k)}, \boldsymbol{w}^{(k)})$ requires computing the expectations in (12) (and may require computing the second order moment of the quantities). In addition, while the problem (10) is strongly concave in $\boldsymbol{w}$, it is potentially *non-convex* in $\boldsymbol\theta$ as the function $V_{\boldsymbol\theta}(\cdot)$ is non-linear with respect to $\boldsymbol\theta \in \Theta$. It is unknown if the primal dual gradient algorithm will converge to a stationary (or saddle) point solution, and if it converges, the rate of convergence is unknown.

## 3  Variance Reduced Policy Evaluation with Nonlinear Approximation

We tackle the policy evaluation problem with smooth function approximation via focusing on a sampled average version of problem (10). To fix idea, we observe a trajectory of state-action pairs $\{s_1, a_1, s_2, a_2, ..., s_m, a_m, s_{m+1}\}$ generated from the policy $\pi$ that we wish to evaluate and consider a sample average approximation of the stochastic objective function (11) as:

$$
\begin{aligned}
\mathcal{L}(\boldsymbol\theta, \boldsymbol{w}) &:= \frac{1}{m}\sum_{i=1}^m \mathcal{L}_i(\boldsymbol\theta, \boldsymbol{w}), \quad \text{where}\\
\mathcal{L}_i(\boldsymbol\theta, \boldsymbol{w}) &:= \Big\langle \boldsymbol{w}, \big[\mathcal{R}(s_i, a_i) + \gamma V_{\boldsymbol\theta}(s_{i+1}) - V_{\boldsymbol\theta}(s_i)\big]g_{\boldsymbol\theta}(s_i)\Big\rangle - \frac{1}{2}(\boldsymbol{w}^\top g_{\boldsymbol\theta}(s_i))^2.
\end{aligned}
\tag{15}
$$

Our goal is to evaluate the stationary point (to be defined later) of the *finite-sum, non-convex, primal-dual* problem:

$$
\min_{\boldsymbol\theta\in\Theta}\max_{\boldsymbol{w}\in\mathsf{W}}\mathcal{L}(\boldsymbol\theta, \boldsymbol{w}) = \frac{1}{m}\sum_{i=1}^m \mathcal{L}_i(\boldsymbol\theta, \boldsymbol{w}).
\tag{16}
$$

**Algorithm 1** Nonconvex Primal-Dual Gradient with Variance Reduction (nPD-VR) Algorithm.

1: **Input**: a trajectory of the state-action pairs $\{s_1, a_1, s_2, a_2, ..., s_m, a_m, s_{m+1}\}$ generated from a given policy; step sizes $\alpha, \beta > 0$; initialization points $\boldsymbol{\theta}^0 \in \Theta$, $\boldsymbol{w}^0 \in \mathbb{R}^d$.

2: Compute the initial averaged gradients as:

$$\mathsf{G}_{\boldsymbol{\theta}}^{(0)} = \tfrac{1}{m} \sum_{i=1}^{m} \nabla_{\boldsymbol{\theta}} \mathcal{L}_i(\boldsymbol{\theta}^{(0)}, \boldsymbol{w}^{(0)}), \quad \mathsf{G}_{\boldsymbol{w}}^{(0)} = \tfrac{1}{m} \sum_{i=1}^{m} \nabla_{\boldsymbol{w}} \mathcal{L}_i(\boldsymbol{\theta}^{(0)}, \boldsymbol{w}^{(0)}) \qquad (18)$$

3: **for** $k = 0, 1, 2, ..., K-1$ **do**

4:     Select two indices $i_k, j_k$ independently and uniformly from $\{1, ..., m\}$.

5:     Perform the primal-dual updates:

$$\boldsymbol{\theta}^{(k+1)} = \mathcal{P}_{\Theta} \left\{ \boldsymbol{\theta}^{(k)} - \beta \Big( \mathsf{G}_{\boldsymbol{\theta}}^{(k)} + \big( \nabla_{\boldsymbol{\theta}} \mathcal{L}_{i_k}(\boldsymbol{\theta}^{(k)}, \boldsymbol{w}^{(k)}) - \nabla_{\boldsymbol{\theta}} \mathcal{L}_{i_k}(\boldsymbol{\theta}_{i_k}^{(k)}, \boldsymbol{w}_{i_k}^{(k)}) \big) \Big) \right\}$$

$$\boldsymbol{w}^{(k+1)} = \boldsymbol{w}^{(k)} + \alpha \Big( \mathsf{G}_{\boldsymbol{w}}^{(k)} + \big( \nabla_{\boldsymbol{w}} \mathcal{L}_{i_k}(\boldsymbol{\theta}^{(k)}, \boldsymbol{w}^{(k)}) - \nabla_{\boldsymbol{w}} \mathcal{L}_{i_k}(\boldsymbol{\theta}_{i_k}^{(k)}, \boldsymbol{w}_{i_k}^{(k)}) \big) \Big),$$

                                                                                   (19)

    where the gradients can be given by (17).

6:     Update the variables as:

$$\boldsymbol{\theta}_i^{(k+1)} = \begin{cases} \boldsymbol{\theta}^{(k)} & \text{if } i = j_k \\ \boldsymbol{\theta}_i^{(k)} & \text{if } i \neq j_k \end{cases}, \quad \boldsymbol{w}_i^{(k+1)} = \begin{cases} \boldsymbol{w}^{(k)} & \text{if } i = j_k \\ \boldsymbol{w}_i^{(k)} & \text{if } i \neq j_k \end{cases} \qquad (20)$$

$$\mathsf{G}_{\boldsymbol{\theta}}^{(k+1)} = \mathsf{G}_{\boldsymbol{\theta}}^{(k)} + \frac{1}{m} \big( \nabla_{\boldsymbol{\theta}} \mathcal{L}_{j_k}(\boldsymbol{\theta}^{(k)}, \boldsymbol{w}^{(k)}) - \nabla_{\boldsymbol{\theta}} \mathcal{L}_{j_k}(\boldsymbol{\theta}_{j_k}^{(k)}, \boldsymbol{w}_{j_k}^{(k)}) \big),$$

$$\mathsf{G}_{\boldsymbol{w}}^{(k+1)} = \mathsf{G}_{\boldsymbol{w}}^{(k)} + \frac{1}{m} \big( \nabla_{\boldsymbol{w}} \mathcal{L}_{j_k}(\boldsymbol{\theta}^{(k)}, \boldsymbol{w}^{(k)}) - \nabla_{\boldsymbol{w}} \mathcal{L}_{j_k}(\boldsymbol{\theta}_{j_k}^{(k)}, \boldsymbol{w}_{j_k}^{(k)}) \big), \qquad (21)$$

7: **end for**

8: **Return**: $(\boldsymbol{\theta}^{(\tilde{K})}, \boldsymbol{w}^{(\tilde{K})})$, where $\tilde{K}$ is independently and uniformly picked from $\{1, ..., K\}$ — an approximate stationary point to (16).

---

Observe that if $m$ is sufficiently large and as $\boldsymbol{G}_{\boldsymbol{\theta}}$ is positive definite, the primal-dual objective function is strongly concave in $\boldsymbol{w}$ but is possibly non-convex in $\boldsymbol{\theta}$ due to non-linearity. The above problem is hence a *one-sided* non-convex problem which remains challenging to tackle.

An exact primal-dual gradient (PDG) algorithm following (14) but replacing the gradients of $L(\boldsymbol{\theta}, \boldsymbol{w})$ by that of $\mathcal{L}(\boldsymbol{\theta}, \boldsymbol{w})$ may be applied to (15). In fact, through exploiting the *one-sided* non-convexity, Lu et al. [24] showed that a similar algorithm to the PDG algorithm indeed converges sublinearly to a stationary point of (16). However, for large $m \gg 1$, implementing the PDG algorithm involves a high per-iteration complexity since evaluating the full gradient requires $\Omega(m)$ FLOPS. Our idea is to derive a fast stochastic algorithm for function approximation through borrowing techniques from variance reduction methods [16, 12, 33, 30].

To fix notations, let $i \in \{1, ..., m\}$ and we define the primal-dual gradient of the $i$th samples:

$$\begin{pmatrix} \nabla_{\boldsymbol{\theta}} \mathcal{L}_i(\boldsymbol{\theta}, \boldsymbol{w}) \\ \nabla_{\boldsymbol{w}} \mathcal{L}_i(\boldsymbol{\theta}, \boldsymbol{w}) \end{pmatrix} = \begin{pmatrix} \big( \delta_i(\boldsymbol{\theta}) - g_{\boldsymbol{\theta}}^{\top}(s_i) \boldsymbol{w} \big) H_{\boldsymbol{\theta}}(s_i) \boldsymbol{w} + \big( g_{\boldsymbol{\theta}}^{\top}(s_i) \boldsymbol{w} \big) \big( \gamma g_{\boldsymbol{\theta}}(s_{i+1}) - g_{\boldsymbol{\theta}}(s_i) \big) \\ \delta_i(\boldsymbol{\theta}) g_{\boldsymbol{\theta}}(s_i) - \big( g_{\boldsymbol{\theta}}(s_i)^{\top} \boldsymbol{w} \big) g_{\boldsymbol{\theta}}(s_i) \end{pmatrix} \qquad (17)$$

where $\delta_i(\boldsymbol{\theta}) := R(s_i, a_i) + \gamma V_{\boldsymbol{\theta}}(s_{i+1}) - V_{\boldsymbol{\theta}}(s_i)$ is the $i$th sampled temporal difference.

We propose the Nonconvex Primal-Dual Gradient with Variance Reduction (nPD-VR) algorithm for (16) in Algorithm 1. The algorithm is a natural extension of the non-convex SAGA algorithm introduced by [30] to the primal-dual, finite-sum setting of interest. In specific, line 5 performs the primal dual gradient update through an unbiased estimate of the gradient — by denoting

$$\widetilde{\mathsf{G}}_{\boldsymbol{\theta}}^{(k)} := \mathsf{G}_{\boldsymbol{\theta}}^{(k)} + \big( \nabla_{\boldsymbol{\theta}} \mathcal{L}_{i_k}(\boldsymbol{\theta}^{(k)}, \boldsymbol{w}^{(k)}) - \nabla_{\boldsymbol{\theta}} \mathcal{L}_{i_k}(\boldsymbol{\theta}_{i_k}^{(k)}, \boldsymbol{w}_{i_k}^{(k)}) \big),$$

$$\widetilde{\mathsf{G}}_{\boldsymbol{w}}^{(k)} := \mathsf{G}_{\boldsymbol{w}}^{(k)} + \big( \nabla_{\boldsymbol{w}} \mathcal{L}_{i_k}(\boldsymbol{\theta}^{(k)}, \boldsymbol{w}^{(k)}) - \nabla_{\boldsymbol{w}} \mathcal{L}_{i_k}(\boldsymbol{\theta}_{i_k}^{(k)}, \boldsymbol{w}_{i_k}^{(k)}) \big), \qquad (22)$$

as $i_k$ is uniformly picked from $\{1, ..., m\}$, therefore (when conditioned on the past random variable generated up to iteration $k$) the expected values of the quantities $\widetilde{\mathsf{G}}_{\boldsymbol{\theta}}^{(k)}, \widetilde{\mathsf{G}}_{\boldsymbol{w}}^{(k)}$ are the primal-dual

gradients $\nabla_{\boldsymbol{\theta}}\mathcal{L}(\boldsymbol{\theta}^{(k)}, \boldsymbol{w}^{(k)})$, $\nabla_{\boldsymbol{w}}\mathcal{L}(\boldsymbol{\theta}^{(k)}, \boldsymbol{w}^{(k)})$, respectively. Meanwhile, the updates in line 6 keep refreshing the stored variables in the memory. We remark that these updates are based on the index $j_k$ which is independent from the $i_k$ used in line 5. As we shall see in the analysis, this subtle detail in the algorithm allows for proving that the variance in gradient is reduced [cf. Lemma 3].

As the nPD-VR algorithm employs an incremental update rule similar to the SAGA method, this algorithm is suitable for the big-data setting when $m \gg 1$. Particularly, the cost for the updates in line 5 and line 6 are independent of $m$. Moreover, the proposed algorithm utilizes a fixed step size rule which allows for adaptation to more dynamical data.

We remark that existing approach [4, 7] have studied a *two-timescale stochastic approximation* algorithm for tackling the stochastic problem (10); and in a similar vein, a recent related work [22] proposed a double loop algorithm that requires solving the dual problem (nearly) optimally. In contrast, the nPD-VR algorithm runs on a *single-timescale*. The nPD-VR algorithm is more flexible and numerically stable, as we shall show in the convergence analysis.

### 3.1 Preliminary Numerical Experiments

We present preliminary experiments of learning the value function from the MountainCar dataset with $m = 5000$ via the nPD-VR algorithm. We ran Sarsa [39] to obtain a good policy, then we generate a trajectory of the state-action pairs. To learn the value function, we parameterize $V_{\boldsymbol{\theta}}(\cdot)$ as a 2-layer neural network with $n$ hidden neurons and consider a forgetting factor $\gamma = 0.95$. We set the constraints in (16) with $\Theta = [0, 1]^n$, and in addition we consider $\boldsymbol{w}$ to be bounded in $[0, 100]^n$ for better numerical stability, which can be enforced

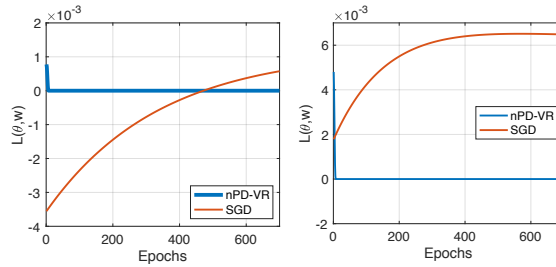

Figure 1: Trajectory of the nPD-VR on the MountainCar dataset such that the value function is approximated as a 2-layer neural network with $n$ neurons. (Left) $n = 50$ neurons (Right) $n = 100$ neurons.

by incorporating a projection step after (19). For the nPD-VR algorithm, we set the step sizes as $\alpha = 10^{-4}$, $\beta = 10^{-8}$. Note we have approximated the Hessian in gradient computation (17) with diagonal approximation. For benchmark, we also experiment with a single-timescale SGD on (16) with a diminishing step size. Trajectory of the objective $\mathcal{L}(\boldsymbol{\theta}^{(k)}, \boldsymbol{w}^{(k)})$ is shown in Fig. 1. As seen, the objective of nPD-VR converges to (close to) zero in 4-5 passes on the data, while a single timescale SGD on (16) takes a long time (or fail) to converge.

## 4 Convergence Analysis

Before stating the main results, let us list a few assumptions on the nPD-VR algorithm and the primal-dual problem (16).

**Assumption 1.** *For any* $\boldsymbol{\theta} \in \Theta$, *the sum function* $\mathcal{L}(\boldsymbol{\theta}, \boldsymbol{w})$ *is* $\mu$*-strongly concave in* $\boldsymbol{w}$.

In the case of policy evaluation problem, Assumption 1 can be implied by taking a sufficient number of samples $m$ and exploiting the fact that $\boldsymbol{G}_{\boldsymbol{\theta}}$ in (7) is positive definite.

**Assumption 2.** *The iterates* $\{\boldsymbol{\theta}^{(k)}, \boldsymbol{w}^{(k)}\}_{k \geq 0}$ *generated by the* nPD-VR *algorithm stay within a compact set* $\Theta \times W$, *for some* $W \subseteq \mathbb{R}^d$ *which is compact and convex.*

Due to the Euclidean projection in the primal-update of $\boldsymbol{\theta}$, the condition $\boldsymbol{\theta}^{(k)} \in \Theta$ holds straightforwardly. Meanwhile it maybe difficult to verify $\boldsymbol{w}^{(k)} \in W$ as the update is unconstrained in general. An intuition is that as $\mathcal{L}(\bar{\boldsymbol{\theta}}, \boldsymbol{w})$ is strongly concave in $\boldsymbol{w}$, for each $\bar{\boldsymbol{\theta}} \in \Theta$, the maximizer to $\mathcal{L}(\bar{\boldsymbol{\theta}}, \boldsymbol{w})$ is unique, *i.e.*, denoted as $\boldsymbol{w}^{\star}(\bar{\boldsymbol{\theta}})$. Also due to the strong concavity, at each iteration $k$ and with a sufficiently small step size, the dual update of $\boldsymbol{w}^k$ pulls the dual variable towards $\boldsymbol{w}^{\star}(\boldsymbol{\theta}^{(k)})$ and therefore $\boldsymbol{w}^k$ also stays within a compact set. Nevertheless, in our numerical experiments in Sec. 3.1, we find that incorporating an additional projection step to the dual update improves the numerical performance. Lastly, we assume that:

**Assumption 3.** *For each $i \in \{1, ..., m\}$, the gradient $\nabla_{\boldsymbol{\theta}} \mathcal{L}_i(\boldsymbol{\theta}, \boldsymbol{w})$ (resp. $\nabla_{\boldsymbol{w}} \mathcal{L}_i(\boldsymbol{\theta}, \boldsymbol{w})$) is $L_{\boldsymbol{\theta}}$ (resp. $L_{\boldsymbol{w}}$) Lipschitz. We have:*

$$\|\nabla_{\boldsymbol{\theta}} \mathcal{L}_i(\boldsymbol{\theta}, \boldsymbol{w}) - \nabla_{\boldsymbol{\theta}} \mathcal{L}_i(\boldsymbol{\theta}', \boldsymbol{w}')\| \leq L_{\boldsymbol{\theta}} (\|\boldsymbol{\theta} - \boldsymbol{\theta}'\| + \|\boldsymbol{w} - \boldsymbol{w}'\|),$$
$$\|\nabla_{\boldsymbol{w}} \mathcal{L}_i(\boldsymbol{\theta}, \boldsymbol{w}) - \nabla_{\boldsymbol{w}} \mathcal{L}_i(\boldsymbol{\theta}', \boldsymbol{w}')\| \leq L_{\boldsymbol{w}} (\|\boldsymbol{\theta} - \boldsymbol{\theta}'\| + \|\boldsymbol{w} - \boldsymbol{w}'\|),$$

(23)

*for any $\boldsymbol{\theta}, \boldsymbol{\theta}' \in \Theta$ and any $\boldsymbol{w}, \boldsymbol{w}' \in \mathsf{W}$, where $\mathsf{W}$ is defined in Assumption 2.*

Assumption 3 is mild and it can be verified by using the compactness of $\mathsf{W}$ and checking (17). In particular, the assumption holds when the parametric family of functions has bounded, smooth gradient and Hessian.

**Summary of Main Results** The primal-dual optimization (16) is a one-sided constrained problem, *i.e.,* only $\boldsymbol{\theta}$ is constrained to $\Theta$ while $\boldsymbol{w}$ is unconstrained. We quantify its convergence via the following stationarity measure. Define $\overline{\boldsymbol{\theta}} = \mathcal{P}_{\Theta}\{\boldsymbol{\theta} - \beta \nabla_{\boldsymbol{\theta}} \mathcal{L}(\boldsymbol{\theta}, \boldsymbol{w})\}$ for any $\boldsymbol{\theta}, \boldsymbol{w} \in \Theta \times \mathbb{R}^d$. Observe that if $\|\overline{\boldsymbol{\theta}} - \boldsymbol{\theta}\| = 0$ and $\nabla_{\boldsymbol{w}} \mathcal{L}(\boldsymbol{\theta}, \boldsymbol{w}) = \mathbf{0}$, then $(\boldsymbol{\theta}, \boldsymbol{w})$ is a (first order) stationary point. Inspired by such observation, the following stationarity measure emerges as a natural metric:

$$\mathcal{G}(\boldsymbol{\theta}^{(k)}, \boldsymbol{w}^{(k)}) := \frac{1}{\beta^2} \|\overline{\boldsymbol{\theta}}^{(k)} - \boldsymbol{\theta}^{(k)}\|^2 + \|\nabla_{\boldsymbol{w}} \mathcal{L}(\boldsymbol{\theta}^{(k)}, \boldsymbol{w}^{(k)})\|^2,$$

(24)

where $\overline{\boldsymbol{\theta}}^{(k)}$ is defined through $(\boldsymbol{\theta}^{(k)}, \boldsymbol{w}^{(k)})$ as

$$\overline{\boldsymbol{\theta}}^{(k)} := \mathcal{P}_{\Theta}\{\boldsymbol{\theta}^{(k)} - \beta \nabla_{\boldsymbol{\theta}} \mathcal{L}(\boldsymbol{\theta}^{(k)}, \boldsymbol{w}^{(k)})\}.$$

(25)

Observe that if $\mathcal{G}(\boldsymbol{\theta}^{(k)}, \boldsymbol{w}^{(k)}) = 0$, then the primal-dual solution $(\boldsymbol{\theta}^{(k)}, \boldsymbol{w}^{(k)})$ is a stationary point. Furthermore, the metric is roughly invariant with the step size since $\|\overline{\boldsymbol{\theta}}^{(k)} - \boldsymbol{\theta}^{(k)}\|^2 = \mathcal{O}(\beta^2)$. The following theorem shows the convergence of the nPD-VR algorithm:

---

**Theorem 1.** *Assume Assumption 1–3 hold true. There exist step size parameters – of the order $\beta = \Theta(1/m), \alpha = \Theta(1/m)$ – such that it holds for any $K \in \mathbb{N}$ that*

$$\mathbb{E}\big[\mathcal{G}(\boldsymbol{\theta}^{(\tilde{K})}, \boldsymbol{w}^{(\tilde{K})})\big] \leq \frac{F^{(K)} + \frac{4}{\mu}\Big(3 + 2m\big(2L_{\boldsymbol{w}}^2 \alpha + L_{\boldsymbol{\theta}}^2 \beta\big)\Big) \|\nabla_{\boldsymbol{w}} \mathcal{L}(\boldsymbol{\theta}^{(0)}, \boldsymbol{w}^{(0)})\|^2}{K \min\{\alpha, \frac{\beta}{4}\}},$$

(26)

*where $F^{(K)} := \mathbb{E}[\mathcal{L}(\boldsymbol{\theta}^{(0)}, \boldsymbol{w}^{(0)}) - \mathcal{L}(\boldsymbol{\theta}^{(K)}, \boldsymbol{w}^{(K)})]$ and we recall that $\tilde{K}$ is a uniform random variable drawn from $\{1, ..., K\}$.*

---

The above shows that the stationarity measure decays to zero at a sublinear rate. In particular, with the step size order $\alpha = \Theta(1/m), \beta = \Theta(1/m)$, the number of iterations required to reach an $\epsilon$-stationary point [with $\mathcal{G}(\boldsymbol{\theta}, \boldsymbol{w}) = \mathcal{O}(\epsilon)$] is $\mathcal{O}(m/\epsilon)$, provided that the strong concavity constant $\mu$, Lipschitz constants of the functions $L_{\boldsymbol{\theta}}, L_{\boldsymbol{w}}$ are independent of $m$.

**Comparison to Prior Work** Note that *non-asymptotic* convergence of primal-dual gradient type algorithms to stationary points with (one sided) *non-convex* problems has only been recently researched. Of close relationship is [24] which study a block coordinate descent version of single loop primal-dual gradient method – the primal and dual updates are performed in sequence and complete gradients are evaluated at each iteration – Lu et al. [24] showed that their algorithm converges to an $\epsilon$-stationary point using $\mathcal{O}(1/\epsilon)$ iterations, under a similar set of assumptions as ours. Since each iteration of [24] requires a complete gradient evaluation, the number of calls to a gradient oracle is thus $\mathcal{O}(m/\epsilon)$. In [22, 29], several proximally guided stochastic mirror descent methods (PG-SMD) are proposed for primal-dual problems following a closely related set of assumptions. However, the PG-SMD methods in [22, 29] rely on a double-loop update in which the primal variables are updated in a faster pace than the dual variables. Nevertheless, [22, 29] show that these methods converges to an $\epsilon$-stationary point using $\mathcal{O}(m/\epsilon)$ gradient oracle calls. To the best of our knowledge, our algorithm is the first stochastic algorithm that can deal with the finite-sum primal-dual problem such as (16), using a single-loop, and variance reduced techniques. Furthermore, the convergence rate of the proposed nPD-VR algorithm is on-par with the state-of-the-art methods.

## 4.1 Proof Outline

Our analysis follows from combining and improving recent techniques for analyzing non-convex optimization algorithms in [30, 24]. To facilitate our analysis, we denote the errors in gradient by $\mathsf{e}_{\boldsymbol{\theta}}^{(k)} := \widetilde{\mathsf{G}}_{\boldsymbol{\theta}}^{(k)} - \nabla_{\boldsymbol{\theta}} \mathcal{L}(\boldsymbol{\theta}^{(k)}, \boldsymbol{w}^{(k)})$ and $\mathsf{e}_{\boldsymbol{w}}^{(k)} := \widetilde{\mathsf{G}}_{\boldsymbol{w}}^{(k)} - \nabla_{\boldsymbol{w}} \mathcal{L}(\boldsymbol{\theta}^{(k)}, \boldsymbol{w}^{(k)})$, respectively. Detailed proofs of results in this section can be found in the supplementary materials.

**Key Lemmas**   We begin by establishing a few lemmas for the convergence analysis. In specific, the first step is to control the change in objective function value with the primal update:

**Lemma 1.** *Under Assumption 3. For any $k \in \mathbb{N}$, we have*

$$\mathcal{L}(\boldsymbol{\theta}^{(k+1)}, \boldsymbol{w}^{(k)}) - \mathcal{L}(\boldsymbol{\theta}^{(k)}, \boldsymbol{w}^{(k)}) \leq \left( L_{\boldsymbol{\theta}} - \frac{1}{2\beta} \right) \|\overline{\boldsymbol{\theta}}^{(k)} - \boldsymbol{\theta}^{(k)}\|^2$$
$$+ \left( \frac{L_{\boldsymbol{\theta}}}{2} - \frac{1}{2\beta} \right) \|\boldsymbol{\theta}^{(k+1)} - \boldsymbol{\theta}^{(k)}\|^2 + \frac{\beta}{2} \|\mathsf{e}_{\boldsymbol{\theta}}^{(k)}\|^2 \qquad (27)$$

The proof follows by the standard descent property of smooth functions combined with the variance controlling technique introduced by [30].

Secondly, the progress made by the dual update obeys the following bounds:

**Lemma 2.** *Under Assumption 1-3. For any $k \in \mathbb{N}$, the change in objective value is bounded as:*

$$\mathcal{L}(\boldsymbol{\theta}^{(k+1)}, \boldsymbol{w}^{(k+1)}) - \mathcal{L}(\boldsymbol{\theta}^{(k+1)}, \boldsymbol{w}^{(k)}) \leq \alpha L_{\boldsymbol{w}}^2 \|\boldsymbol{\theta}^{(k+1)} - \boldsymbol{\theta}^{(k)}\|^2$$
$$\left( 2\alpha + \frac{\alpha^3 L_{\boldsymbol{w}}^2}{2} - \frac{\mu\alpha^2}{2} \right) \|\nabla_{\boldsymbol{w}} \mathcal{L}(\boldsymbol{\theta}^{(k)}, \boldsymbol{w}^{(k)})\|^2 + \left( \alpha - \frac{\mu\alpha^2}{2} \right) \|\mathsf{e}_{\boldsymbol{w}}^{(k)}\|^2, \qquad (28)$$

*and the dual gradient is controlled by:*

$$\|\nabla_{\boldsymbol{w}} \mathcal{L}(\boldsymbol{\theta}^{(k)}, \boldsymbol{w}^{(k)})\|^2 \leq \left( 1 + \alpha^2 L_y^2 - 2\mu\alpha \right) \|\nabla_{\boldsymbol{w}} \mathcal{L}(\boldsymbol{\theta}^{(k-1)}, \boldsymbol{w}^{(k-1)})\|^2$$
$$+ \mu\alpha \|\nabla_{\boldsymbol{w}} \mathcal{L}(\boldsymbol{\theta}^{(k)}, \boldsymbol{w}^{(k)})\|^2 + \frac{L_{\boldsymbol{w}}^2}{\mu\alpha} \left( \|\boldsymbol{\theta}^{(k)} - \boldsymbol{\theta}^{(k-1)}\|^2 + \alpha^2 \|\mathsf{e}_{\boldsymbol{w}}^{(k-1)}\|^2 \right) \qquad (29)$$

The bound (28) is a standard relation for dual gradient update, while (29) is a consequence of the strong-concavity of $\mathcal{L}(\boldsymbol{\theta}^{(k)}, \boldsymbol{w}^{(k)})$ – it shows that $\|\nabla_{\boldsymbol{w}} \mathcal{L}(\boldsymbol{\theta}^{(k)}, \boldsymbol{w}^{(k)})\|$ contracts after a dual update.

To control the gradient error terms in expectation $\|\mathsf{e}_{\boldsymbol{\theta}}^{(k)}\|^2$, $\|\mathsf{e}_{\boldsymbol{w}}^{(k)}\|^2$, we consider [also see Lemma 4 in the supplementary materials]

$$\Delta^{(k)} := \frac{1}{m} \sum_{i=1}^{m} \left\{ \|\boldsymbol{\theta}^{(k)} - \boldsymbol{\theta}_i^{(k)}\|^2 + \|\boldsymbol{w}^{(k)} - \boldsymbol{w}_i^{(k)}\|^2 \right\} \qquad (30)$$

and notice that $\Delta^{(0)} = 0$. Using Assumption 3 and when the step size is sufficiently small, we can establish a bound on $\sum_{k=0}^{K} \mathbb{E}[\Delta^{(k)}]$ via the below lemma:

**Lemma 3.** *Under Assumption 3 and the condition on the step sizes that:*

$$\delta(\alpha, \beta) := \frac{1}{m} - \max\{\alpha, \beta\} - 4L_{\boldsymbol{w}}^2 \left( \alpha^2 + \alpha(1 - \frac{1}{m}) \right) > 0. \qquad (31)$$

*For any $K \geq 1$, we have*

$$\sum_{k=0}^{K} \mathbb{E}[\Delta^{(k)}] \leq \frac{1}{\delta(\alpha, \beta)} \sum_{k=0}^{K} \mathbb{E}\left\{ \frac{2}{\beta} \|\boldsymbol{\theta}^{(k+1)} - \boldsymbol{\theta}^{(k)}\|^2 + 4\alpha \|\nabla_{\boldsymbol{w}} \mathcal{L}(\boldsymbol{\theta}^{(k)}, \boldsymbol{w}^{(k)})\|^2 \right\}. \qquad (32)$$

The proof of the lemma makes use of the property of the nPD-VR algorithm and uses a new technique in proving the contraction of variance in SAGA-type algorithms. Furthermore, note that if the step sizes satisfies

$$\frac{1}{2m} \geq \max\{\alpha, \beta\} + 8\alpha L_{\boldsymbol{w}}^2, \qquad \text{(a0)}$$

then one has $(\frac{1}{m} - \max\{\alpha, \beta\} - 4L_{\boldsymbol{w}}^2(\alpha^2 + \alpha(1 - \frac{1}{m})))^{-1} \leq \frac{m}{2}$. We simplify (32) into

$$\sum_{k=0}^{K} \mathbb{E}[\Delta^{(k)}] \leq \sum_{k=0}^{K} \mathbb{E}\left\{ \frac{m}{\beta} \|\boldsymbol{\theta}^{(k+1)} - \boldsymbol{\theta}^{(k)}\|^2 + 2m\alpha \|\nabla_{\boldsymbol{w}} \mathcal{L}(\boldsymbol{\theta}^{(k)}, \boldsymbol{w}^{(k)})\|^2 \right\}. \qquad (33)$$

**Proof of Theorem 1** Equipped with the lemmas above on the progress made by primal-dual updates and the SAGA gradient estimation, our proof follows by analyzing (27), (28), (29). We remark that the proof technique used is new, which departs from the common Lyapunov/potential function approach pursued in recent papers [30, 24] on non-convex analysis.

To illustrate the idea, through carefully controlling the step size, we show that by summing up the inequalities (27), (28) from $k = 0$ to $k = K - 1$, we get

$$\Omega\big(\min\{\alpha, \beta\}\big) \sum_{k=0}^{K-1} \mathbb{E}[\mathcal{G}(\boldsymbol{\theta}^{(k)}, \boldsymbol{w}^{(k)})] \leq \mathcal{O}(\alpha) \sum_{k=0}^{K-1} \mathbb{E}[\|\nabla_{\boldsymbol{w}} \mathcal{L}(\boldsymbol{\theta}^{(k)}, \boldsymbol{w}^{(k)})\|^2]$$
$$+ \mathcal{O}(m - \tfrac{1}{\beta}) \sum_{k=0}^{K-1} \mathbb{E}[\|\boldsymbol{\theta}^{(k+1)} - \boldsymbol{\theta}^{(k)}\|^2] + \text{constant.} \tag{34}$$

Using (29), the sum $\sum_{k=0}^{K-1} \mathbb{E}[\|\nabla_{\boldsymbol{w}} \mathcal{L}(\boldsymbol{\theta}^{(k)}, \boldsymbol{w}^{(k)})\|^2]$ can be further upper bounded as the form constant $\times \sum_{k=0}^{K-1} \mathbb{E}[\|\boldsymbol{\theta}^{(k+1)} - \boldsymbol{\theta}^{(k)}\|^2] + \text{constant}$. Substituting the newly obtained bound, one can find a step size $\beta > 0$ such that the constant in front of the term $\mathbb{E}[\|\boldsymbol{\theta}^{(k+1)} - \boldsymbol{\theta}^{(k)}\|^2]$ is negative. It follows that we can upper bound the right hand side of (34) with a constant independent of $K$.

Subsequently, we observe that as $\tilde{K}$ is an independent r.v. uniformly distributed on $\{0, ..., K - 1\}$, one has $\mathbb{E}\big[\mathcal{G}(\boldsymbol{\theta}^{(\tilde{K})}, \boldsymbol{w}^{(\tilde{K})})\big] = K^{-1} \sum_{k=0}^{K-1} \mathbb{E}[\mathcal{G}(\boldsymbol{\theta}^{(k)}, \boldsymbol{w}^{(k)})]$ and applying (34) yields Theorem 1.

## 5   Conclusions and Extensions

In this paper, we have studied the policy evaluation problem in the case of smooth (possibly non-linear) function approximation. We consider an offline setting via sample average approximation of the Bellman equation. Albeit the sample size $m$ can be large, we propose a simple and efficient, variance reduced primal dual update strategy to handle the one-sided non-convex optimization problem arisen. We analyze the non-asymptotic convergence rate of the algorithm towards a stationary point and demonstrate that it performs on par with state-of-the-art optimization methods, while the latter requires higher implementation complexity.

Several extensions are worth studying — similar to the SAGA algorithm considered here, the SVRG algorithm [16] may benefit the nonconvex primal-dual optimization; as suggested by [30], using mini-batch can accelerate the convergence rate from $\mathcal{O}(m/K)$ to $\mathcal{O}(m^{\frac{2}{3}}/K)$.

## Acknowledgement

H.-T. Wai is supported by the CUHK Direct Grant #4055113. M. Hong is supported in part by NSF under Grant CCF-1651825, CMMI-172775, CIF-1910385 and by AFOSR under grant 19RT0424.

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
