[Supplementary Material]

# A  Proof of Lemma 1

**Lemma.** *Under Assumption 3. For any $k \in \mathbb{N}$, we have*

$$\mathcal{L}(\boldsymbol{\theta}^{(k+1)}, \boldsymbol{w}^{(k)}) - \mathcal{L}(\boldsymbol{\theta}^{(k)}, \boldsymbol{w}^{(k)}) \leq \left( L_{\boldsymbol{\theta}} - \frac{1}{2\beta} \right) \|\overline{\boldsymbol{\theta}}^{(k)} - \boldsymbol{\theta}^{(k)}\|^2 \qquad (35)$$
$$+ \left( \frac{L_{\boldsymbol{\theta}}}{2} - \frac{1}{2\beta} \right) \|\boldsymbol{\theta}^{(k+1)} - \boldsymbol{\theta}^{(k)}\|^2 + \frac{\beta}{2} \|\mathsf{e}_{\boldsymbol{\theta}}^{(k)}\|^2$$

*Proof.* First, using the smoothness of $\mathcal{L}$, we observe that

$$\mathcal{L}(\overline{\boldsymbol{\theta}}^{(k)}, \boldsymbol{w}^{(k)}) - f(\boldsymbol{\theta}^{(k)}, \boldsymbol{w}^{(k)}) \leq \langle \nabla_{\boldsymbol{\theta}} \mathcal{L}(\boldsymbol{\theta}^{(k)}, \boldsymbol{w}^{(k)}), \overline{\boldsymbol{\theta}}^{(k)} - \boldsymbol{\theta}^{(k)} \rangle + \frac{L_{\boldsymbol{\theta}}}{2} \|\overline{\boldsymbol{\theta}}^{(k)} - \boldsymbol{\theta}^{(k)}\|^2 \qquad (36)$$
$$\leq \left( \frac{L_{\boldsymbol{\theta}}}{2} - \frac{1}{\beta} \right) \|\overline{\boldsymbol{\theta}}^{(k)} - \boldsymbol{\theta}^{(k)}\|^2$$

where the last inequality uses the optimality of $\overline{\boldsymbol{\theta}}^{(k)}$ which upper bounds the inner product by zero. Second, applying Lemma 2 from [30], we have

$$\mathcal{L}(\boldsymbol{\theta}^{(k+1)}, \boldsymbol{w}^{(k)}) - \mathcal{L}(\overline{\boldsymbol{\theta}}^{(k)}, \boldsymbol{w}^{(k)}) \leq \langle \boldsymbol{\theta}^{(k+1)} - \overline{\boldsymbol{\theta}}^{(k)}, \nabla_{\boldsymbol{\theta}} \mathcal{L}(\boldsymbol{\theta}^{(k)}, \boldsymbol{w}^{(k)}) - \widetilde{\mathsf{G}}_{\boldsymbol{\theta}}^{(k)} \rangle$$
$$\left( \frac{L_{\boldsymbol{\theta}}}{2} - \frac{1}{2\beta} \right) \|\boldsymbol{\theta}^{(k+1)} - \boldsymbol{\theta}^{(k)}\|^2 + \left( \frac{L_{\boldsymbol{\theta}}}{2} + \frac{1}{2\beta} \right) \|\overline{\boldsymbol{\theta}}^{(k)} - \boldsymbol{\theta}^{(k)}\|^2 - \frac{1}{2\beta} \|\boldsymbol{\theta}^{(k+1)} - \overline{\boldsymbol{\theta}}^{(k)}\|^2 \qquad (37)$$

Adding the two inequalities together gives

$$\mathcal{L}(\boldsymbol{\theta}^{(k+1)}, \boldsymbol{w}^{(k)}) - \mathcal{L}(\boldsymbol{\theta}^{(k)}, \boldsymbol{w}^{(k)}) \leq \langle \boldsymbol{\theta}^{(k+1)} - \overline{\boldsymbol{\theta}}^{(k)}, \nabla_{\boldsymbol{\theta}} \mathcal{L}(\boldsymbol{\theta}^{(k)}, \boldsymbol{w}^{(k)}) - \widetilde{\mathsf{G}}_{\boldsymbol{\theta}}^{(k)} \rangle$$
$$\left( \frac{L_{\boldsymbol{\theta}}}{2} - \frac{1}{2\beta} \right) \|\boldsymbol{\theta}^{(k+1)} - \boldsymbol{\theta}^{(k)}\|^2 + \left( L_{\boldsymbol{\theta}} - \frac{1}{2\beta} \right) \|\overline{\boldsymbol{\theta}}^{(k)} - \boldsymbol{\theta}^{(k)}\|^2 - \frac{1}{2\beta} \|\boldsymbol{\theta}^{(k+1)} - \overline{\boldsymbol{\theta}}^{(k)}\|^2 \qquad (38)$$

Noting that by the Young's inequality, one has

$$\langle \boldsymbol{\theta}^{(k+1)} - \overline{\boldsymbol{\theta}}^{(k)}, \nabla_{\boldsymbol{\theta}} \mathcal{L}(\boldsymbol{\theta}^{(k)}, \boldsymbol{w}^{(k)}) - \widetilde{\mathsf{G}}_{\boldsymbol{\theta}}^{(k)} \rangle$$
$$\leq \frac{1}{2\beta} \|\boldsymbol{\theta}^{(k+1)} - \overline{\boldsymbol{\theta}}^{(k)}\|^2 + \frac{\beta}{2} \|\nabla_{\boldsymbol{\theta}} \mathcal{L}(\boldsymbol{\theta}^{(k)}, \boldsymbol{w}^{(k)}) - \widetilde{\mathsf{G}}_{\boldsymbol{\theta}}^{(k)}\|^2 \qquad (39)$$

Noting that $\nabla_{\boldsymbol{\theta}} \mathcal{L}(\boldsymbol{\theta}^{(k)}, \boldsymbol{w}^{(k)}) - \widetilde{\mathsf{G}}_{\boldsymbol{\theta}}^{(k)} = -\mathsf{e}_{\boldsymbol{\theta}}^{(k)}$, substituting back into (38) yields our claim. $\qquad \square$

# B  Proof of Lemma 2

**Lemma.** *Under Assumption 1-3. For any $k \in \mathbb{N}$, the change in objective value is bounded as:*

$$\mathcal{L}(\boldsymbol{\theta}^{(k+1)}, \boldsymbol{w}^{(k+1)}) - \mathcal{L}(\boldsymbol{\theta}^{(k+1)}, \boldsymbol{w}^{(k)}) \leq \alpha L_{\boldsymbol{w}}^2 \|\boldsymbol{\theta}^{(k+1)} - \boldsymbol{\theta}^{(k)}\|^2$$
$$\left( 2\alpha + \frac{\alpha^3 L_{\boldsymbol{w}}^2}{2} - \frac{\mu \alpha^2}{2} \right) \|\nabla_{\boldsymbol{w}} \mathcal{L}(\boldsymbol{\theta}^{(k)}, \boldsymbol{w}^{(k)})\|^2 + \left( \alpha - \frac{\mu \alpha^2}{2} \right) \|\mathsf{e}_{\boldsymbol{w}}^{(k)}\|^2, \qquad (40)$$

*and the dual gradient is controlled by:*

$$\|\nabla_{\boldsymbol{w}} \mathcal{L}(\boldsymbol{\theta}^{(k)}, \boldsymbol{w}^{(k)})\|^2 \leq \left( 1 + \alpha^2 L_y^2 - 2\mu\alpha \right) \|\nabla_{\boldsymbol{w}} \mathcal{L}(\boldsymbol{\theta}^{(k-1)}, \boldsymbol{w}^{(k-1)})\|^2$$
$$+ \mu\alpha \|\nabla_{\boldsymbol{w}} \mathcal{L}(\boldsymbol{\theta}^{(k)}, \boldsymbol{w}^{(k)})\|^2 + \frac{L_y^2}{\mu\alpha} \left( \|\boldsymbol{\theta}^{(k)} - \boldsymbol{\theta}^{(k-1)}\|^2 + \alpha^2 \|\mathsf{e}_{\boldsymbol{w}}^{(k-1)}\|^2 \right) \qquad (41)$$

*Proof.* Let us define $\overline{\boldsymbol{w}}^{(k)} = \boldsymbol{w}^{(k)} + \alpha \nabla_{\boldsymbol{w}} \mathcal{L}(\boldsymbol{\theta}^{(k)}, \boldsymbol{w}^{(k)})$ as the iterate obtained if the dual update is performed using the exact dual gradient. Observe

$$\nabla_{\boldsymbol{w}} \mathcal{L}(\boldsymbol{\theta}^{(k)}, \boldsymbol{w}^{(k)}) = \frac{1}{\alpha}(\overline{\boldsymbol{w}}^{(k)} - \boldsymbol{w}^{(k)}), \quad \widetilde{\mathsf{G}}_{\boldsymbol{w}}^{(k)} = \frac{1}{\alpha}(\boldsymbol{w}^{(k+1)} - \boldsymbol{w}^{(k)}), \qquad (42)$$

From the above definition, and the definition of $\mathrm{e}_w^{(k)}$ we have

$$\mathrm{e}_w^{(k)} = \widetilde{\mathsf{G}}_w^{(k)} - \nabla_w \mathcal{L}(\boldsymbol{\theta}^{(k)}, \boldsymbol{w}^{(k)}) = \frac{1}{\alpha}(\boldsymbol{w}^{(k+1)} - \overline{\boldsymbol{w}}^{(k)}). \tag{43}$$

We upper bound the difference in objective value by observing that

$$\mathcal{L}(\boldsymbol{\theta}^{(k+1)}, \overline{\boldsymbol{w}}^{(k)}) - \mathcal{L}(\boldsymbol{\theta}^{(k+1)}, \boldsymbol{w}^{(k)}) \le \langle \nabla_w \mathcal{L}(\boldsymbol{\theta}^{(k+1)}, \boldsymbol{w}^{(k)}), \overline{\boldsymbol{w}}^{(k)} - \boldsymbol{w}^{(k)} \rangle - \frac{\mu}{2}\|\overline{\boldsymbol{w}}^{(k)} - \boldsymbol{w}^{(k)}\|^2$$

$$\mathcal{L}(\boldsymbol{\theta}^{(k+1)}, \boldsymbol{w}^{(k+1)}) - \mathcal{L}(\boldsymbol{\theta}^{(k+1)}, \overline{\boldsymbol{w}}^{(k)}) \le \langle \nabla_w \mathcal{L}(\boldsymbol{\theta}^{(k+1)}, \overline{\boldsymbol{w}}^{(k)}), \boldsymbol{w}^{(k+1)} - \overline{\boldsymbol{w}}^{(k)} \rangle - \frac{\mu}{2}\|\boldsymbol{w}^{(k+1)} - \overline{\boldsymbol{w}}^{(k)}\|^2$$

$$\tag{44}$$

where we have used the $\mu$-strong concavity of $\mathcal{L}$ *w.r.t.* $\boldsymbol{w}$ [cf. Assumption 1]. Adding up the two inequalities and applying the Young's inequality to combine relevant terms yield that

$$\mathcal{L}(\boldsymbol{\theta}^{(k+1)}, \boldsymbol{w}^{(k+1)}) - \mathcal{L}(\boldsymbol{\theta}^{(k+1)}, \boldsymbol{w}^{(k)})$$
$$\le -\frac{\mu}{2}\big(\|\overline{\boldsymbol{w}}^{(k)} - \boldsymbol{w}^{(k)}\|^2 + \|\boldsymbol{w}^{(k+1)} - \overline{\boldsymbol{w}}^{(k)}\|^2\big)$$
$$+ \langle \nabla_w \mathcal{L}(\boldsymbol{\theta}^{(k)}, \boldsymbol{w}^{(k)}), \overline{\boldsymbol{w}}^{(k)} - \boldsymbol{w}^{(k)} \rangle + \langle \nabla_w \mathcal{L}(\boldsymbol{\theta}^{(k)}, \boldsymbol{w}^{(k)}), \boldsymbol{w}^{(k+1)} - \overline{\boldsymbol{w}}^{(k)} \rangle$$
$$+ \frac{1}{2\alpha}\big(\|\overline{\boldsymbol{w}}^{(k)} - \boldsymbol{w}^{(k)}\|^2 + \|\boldsymbol{w}^{(k+1)} - \overline{\boldsymbol{w}}^{(k)}\|^2\big) + \frac{\alpha L_w^2}{2}\big(2\|\boldsymbol{\theta}^{(k+1)} - \boldsymbol{\theta}^{(k)}\|^2 + \|\overline{\boldsymbol{w}}^{(k)} - \boldsymbol{w}^{(k)}\|^2\big) \tag{45}$$

Using the conditions in (42) yields

$$\mathcal{L}(\boldsymbol{\theta}^{(k+1)}, \boldsymbol{w}^{(k+1)}) - \mathcal{L}(\boldsymbol{\theta}^{(k+1)}, \boldsymbol{w}^{(k)}) \le \Big(2\alpha + \frac{\alpha^3 L_w^2}{2} - \frac{\mu\alpha^2}{2}\Big)\|\nabla_w \mathcal{L}(\boldsymbol{\theta}^{(k)}, \boldsymbol{w}^{(k)})\|^2$$
$$+ \Big(\alpha - \frac{\mu\alpha^2}{2}\Big)\|\mathrm{e}_w^{(k)}\|^2 + \alpha L_w^2\|\boldsymbol{\theta}^{(k+1)} - \boldsymbol{\theta}^{(k)}\|^2 \tag{46}$$

This proves the first part in the lemma.

For the second part of the lemma, we observe the equality by subtracting the optimality condition for the two consecutive $\boldsymbol{w}$ problems

$$\frac{1}{\alpha}\big(\underbrace{(\overline{\boldsymbol{w}}^{(k)} - \boldsymbol{w}^{(k)}) - (\overline{\boldsymbol{w}}^{(k-1)} - \boldsymbol{w}^{(k-1)})}_{:=\bar{v}^{(k)}}\big) = \nabla_w \mathcal{L}(\boldsymbol{\theta}^{(k)}, \boldsymbol{w}^{(k)}) - \nabla_w \mathcal{L}(\boldsymbol{\theta}^{(k-1)}, \boldsymbol{w}^{(k-1)}) \tag{47}$$

Taking the inner product with $\overline{\boldsymbol{w}}^{(k)} - \boldsymbol{w}^{(k)}$. The left hand side equals

$$\frac{1}{\alpha}\langle \bar{v}^{(k)}, \overline{\boldsymbol{w}}^{(k)} - \boldsymbol{w}^{(k)} \rangle = \frac{1}{2\alpha}\big(\|\overline{\boldsymbol{w}}^{(k)} - \boldsymbol{w}^{(k)}\|^2 + \|\bar{v}^{(k)}\|^2 - \|\overline{\boldsymbol{w}}^{(k-1)} - \boldsymbol{w}^{(k-1)}\|^2\big) \tag{48}$$

and the right hand side can be upper bounded as

$$\langle \nabla_w \mathcal{L}(\boldsymbol{\theta}^{(k)}, \boldsymbol{w}^{(k)}) - \nabla_w \mathcal{L}(\boldsymbol{\theta}^{(k-1)}, \boldsymbol{w}^{(k-1)}), \overline{\boldsymbol{w}}^{(k)} - \boldsymbol{w}^{(k)} \rangle$$
$$\le \frac{\mu}{2}\|\overline{\boldsymbol{w}}^{(k)} - \boldsymbol{w}^{(k)}\|^2 + \frac{1}{2\mu}\|\nabla_w \mathcal{L}(\boldsymbol{\theta}^{(k)}, \boldsymbol{w}^{(k)}) - \nabla_w \mathcal{L}(\boldsymbol{\theta}^{(k-1)}, \overline{\boldsymbol{w}}^{(k-1)})\|^2$$
$$+ \langle \nabla_w \mathcal{L}(\boldsymbol{\theta}^{(k-1)}, \overline{\boldsymbol{w}}^{(k-1)}) - \nabla_w \mathcal{L}(\boldsymbol{\theta}^{(k-1)}, \boldsymbol{w}^{(k-1)}), \bar{v}^{(k)} + \overline{\boldsymbol{w}}^{(k-1)} - \boldsymbol{w}^{(k-1)} \rangle$$
$$\le \frac{\mu}{2}\|\overline{\boldsymbol{w}}^{(k)} - \boldsymbol{w}^{(k)}\|^2 + \frac{L_w^2}{2\mu}\big(\|\boldsymbol{\theta}^{(k)} - \boldsymbol{\theta}^{(k-1)}\|^2 + \|\boldsymbol{w}^{(k)} - \overline{\boldsymbol{w}}^{(k-1)}\|^2\big)$$
$$+ \frac{1}{2\alpha}\|\bar{v}^{(k)}\|^2 + \Big(-\mu + \frac{\alpha L_w^2}{2}\Big)\|\overline{\boldsymbol{w}}^{(k-1)} - \boldsymbol{w}^{(k-1)}\|^2$$

The above can be simplified as:

$$\frac{1}{2\alpha}\|\overline{\boldsymbol{w}}^{(k)} - \boldsymbol{w}^{(k)}\|^2 \le \Big(\frac{1}{2\alpha} + \frac{\alpha L_w^2}{2} - \mu\Big)\|\overline{\boldsymbol{w}}^{(k-1)} - \boldsymbol{w}^{(k-1)}\|^2 + \frac{\mu}{2}\|\overline{\boldsymbol{w}}^{(k)} - \boldsymbol{w}^{(k)}\|^2$$
$$+ \frac{L_w^2}{2\mu}\big(\|\boldsymbol{\theta}^{(k)} - \boldsymbol{\theta}^{(k-1)}\|^2 + \|\boldsymbol{w}^{(k)} - \overline{\boldsymbol{w}}^{(k-1)}\|^2\big) \tag{49}$$

Substituting $\overline{\boldsymbol{w}}^{(k)} - \boldsymbol{w}^{(k)} = \alpha \nabla_w \mathcal{L}(\boldsymbol{\theta}^{(k)}, \boldsymbol{w}^{(k)})$ yields the desired results. $\qquad \square$

## C  Proof of Lemma 3

**Lemma.** *Under Assumption 3 and the condition on the step sizes that:*

$$\delta(\alpha, \beta) := \frac{1}{m} - \max\{\alpha, \beta\} - 2L_{\boldsymbol{w}}^2(\alpha^2 + \alpha(1 - \frac{1}{m})) > 0. \tag{50}$$

*For any $K \geq 1$, we have*

$$\sum_{k=0}^{K} \mathbb{E}[\Delta^{(k)}] \leq \frac{1}{\delta(\alpha, \beta)} \sum_{k=0}^{K} \mathbb{E}\Big\{\frac{2}{\beta}\|\boldsymbol{\theta}^{(k+1)} - \boldsymbol{\theta}^{(k)}\|^2 + 4\alpha\|\nabla_{\boldsymbol{w}}\mathcal{L}(\boldsymbol{\theta}^{(k)}, \boldsymbol{w}^{(k)})\|^2\Big\}. \tag{51}$$

*Proof.* We begin the proof by observing the following chain:

$$\mathbb{E}\Big[\frac{1}{m}\sum_{i=1}^{m}\|\boldsymbol{\theta}^{(k+1)} - \boldsymbol{\theta}_i^{(k+1)}\|^2\Big]$$

$$\stackrel{(a)}{=} \frac{1}{m}\sum_{i=1}^{m}\mathbb{E}\Big[\frac{1}{m}\|\boldsymbol{\theta}^{(k+1)} - \boldsymbol{\theta}^{(k)}\|^2 + \big(1 - \frac{1}{m}\big)\|\boldsymbol{\theta}^{(k+1)} - \boldsymbol{\theta}_i^{(k)}\|^2\Big]$$

$$\stackrel{(b)}{=} \frac{1}{m}\sum_{i=1}^{m}\mathbb{E}\Big[\|\boldsymbol{\theta}^{(k+1)} - \boldsymbol{\theta}^{(k)}\|^2 + \big(1 - \frac{1}{m}\big)\big(\|\boldsymbol{\theta}^{(k)} - \boldsymbol{\theta}_i^{(k)}\|^2 + 2\langle\boldsymbol{\theta}^{(k+1)} - \boldsymbol{\theta}^{(k)}, \boldsymbol{\theta}^{(k)} - \boldsymbol{\theta}_i^{(k)}\rangle\big)\Big]$$

$$\stackrel{(c)}{\leq} \frac{1}{m}\sum_{i=1}^{m}\mathbb{E}\Big[\big(1 + \frac{1 - \frac{1}{m}}{\beta}\big)\|\boldsymbol{\theta}^{(k+1)} - \boldsymbol{\theta}^{(k)}\|^2 + \big(1 - \frac{1}{m}\big)(1 + \beta)\|\boldsymbol{\theta}^{(k)} - \boldsymbol{\theta}_i^{(k)}\|^2\Big] \tag{52}$$

where (a) uses the fact that $j_k$ is uniformly picked from $\{1, ..., m\}$ and is independent from the index $i_k$ that is used for updating $\boldsymbol{\theta}^{(k+1)}$, (b) uses the decomposition $\boldsymbol{\theta}^{(k+1)} - \boldsymbol{\theta}_i^{(k)} = \boldsymbol{\theta}^{(k+1)} - \boldsymbol{\theta}^{(k)} + \boldsymbol{\theta}^{(k)} - \boldsymbol{\theta}_i^{(k)}$ and expanding the quadratic term, (c) is due to Young's inequality. Similarly the above reasoning can be applied to the dual variables to yield

$$\mathbb{E}\Big[\frac{1}{m}\sum_{i=1}^{m}\|\boldsymbol{w}^{(k+1)} - \boldsymbol{w}_i^{(k+1)}\|^2\Big]$$

$$\leq \frac{1}{m}\sum_{i=1}^{m}\mathbb{E}\Big[\big(1 + \frac{1 - \frac{1}{m}}{\alpha}\big)\|\boldsymbol{w}^{(k+1)} - \boldsymbol{w}^{(k)}\|^2 + \big(1 - \frac{1}{m}\big)(1 + \alpha)\|\boldsymbol{w}^{(k)} - \boldsymbol{w}_i^{(k)}\|^2\Big] \tag{53}$$

Combining the above shows that

$$\mathbb{E}[\Delta^{(k+1)}] \leq \big(1 + \frac{1 - \frac{1}{m}}{\beta}\big)\|\boldsymbol{\theta}^{(k+1)} - \boldsymbol{\theta}^{(k)}\|^2 + \big(1 + \frac{1 - \frac{1}{m}}{\alpha}\big)\|\boldsymbol{w}^{(k+1)} - \boldsymbol{w}^{(k)}\|^2$$

$$+ \big(1 - \frac{1}{m}\big)\max\{1 + \alpha, 1 + \beta\}\mathbb{E}[\Delta^{(k)}]$$

$$\leq \big(1 + \frac{1 - \frac{1}{m}}{\beta}\big)\|\boldsymbol{\theta}^{(k+1)} - \boldsymbol{\theta}^{(k)}\|^2 + \big(1 + \frac{1 - \frac{1}{m}}{\alpha}\big)\|\boldsymbol{w}^{(k+1)} - \boldsymbol{w}^{(k)}\|^2 \tag{54}$$

$$+ \big(1 - \frac{1}{m} + \max\{\alpha, \beta\}\big)\mathbb{E}[\Delta^{(k)}]$$

Telescoping with the above term gives the geometric series

$$\mathbb{E}[\Delta^{(k+1)}] \leq \sum_{\ell=0}^{k}\big(1 - \frac{1}{m} + \max\{\alpha, \beta\}\big)^{k-\ell}\Big\{\big(1 + \frac{1 - \frac{1}{m}}{\beta}\big)\|\boldsymbol{\theta}^{(\ell+1)} - \boldsymbol{\theta}^{(\ell)}\|^2 + \big(1 + \frac{1 - \frac{1}{m}}{\alpha}\big)\|\boldsymbol{w}^{(\ell+1)} - \boldsymbol{w}^{(\ell)}\|^2\Big\}$$

For $\frac{1}{m} > \max\{\alpha, \beta\}$, it can be easily derived that

$$\sum_{k=0}^{K}\mathbb{E}[\Delta^{(k)}] \leq \sum_{k=0}^{K}\frac{1}{\frac{1}{m} - \max\{\alpha, \beta\}}\Big\{\big(1 + \frac{1 - \frac{1}{m}}{\beta}\big)\|\boldsymbol{\theta}^{(k+1)} - \boldsymbol{\theta}^{(k)}\|^2 + \big(1 + \frac{1 - \frac{1}{m}}{\alpha}\big)\|\boldsymbol{w}^{(k+1)} - \boldsymbol{w}^{(k)}\|^2\Big\} \tag{55}$$

Further upper bounding $\|\boldsymbol{w}^{(k+1)} - \boldsymbol{w}^{(k)}\|^2$ by $2\big(\alpha^2\|\mathsf{e}_{\boldsymbol{w}}^{(k)}\|^2 + \|\boldsymbol{w}^{(k)} - \overline{\boldsymbol{w}}^{(k)}\|^2\big)$ [cf. Young's inequality], and by utilizing Lemma 4 yield

$$\sum_{k=0}^{K}\mathbb{E}\big[\Delta^{(k)}\big] \leq$$

$$\sum_{k=0}^{K}\frac{1}{\frac{1}{m} - \max\{\alpha,\beta\}}\left\{\left(1 + \frac{1 - \frac{1}{m}}{\beta}\right)\|\boldsymbol{\theta}^{(k+1)} - \boldsymbol{\theta}^{(k)}\|^2 + \left(2 + \frac{2 - \frac{2}{m}}{\alpha}\right)\big(\|\overline{\boldsymbol{w}}^{(k)} - \boldsymbol{w}^{(k)}\|^2 + 2\alpha^2 L_{\boldsymbol{w}}^2\Delta^{(k)}\big)\right\} \tag{56}$$

Reshuffling the terms related to $\sum_{k=0}^{K}\mathbb{E}\big[\Delta^{(k)}\big]$ and recalling the definition of $\delta(\alpha,\beta)$ yields that

$$\sum_{k=0}^{K}\mathbb{E}\big[\Delta^{(k)}\big] \leq$$

$$\sum_{k=0}^{K}\frac{1}{\delta(\alpha,\beta)}\left\{\left(1 + \frac{1 - \frac{1}{m}}{\beta}\right)\|\boldsymbol{\theta}^{(k+1)} - \boldsymbol{\theta}^{(k)}\|^2 + \left(2 + \frac{2 - \frac{2}{m}}{\alpha}\right)\|\overline{\boldsymbol{w}}^{(k)} - \boldsymbol{w}^{(k)}\|^2\right\} \tag{57}$$

$$\sum_{k=0}^{K}\frac{1}{\delta(\alpha,\beta)}\left\{\frac{2}{\beta}\|\boldsymbol{\theta}^{(k+1)} - \boldsymbol{\theta}^{(k)}\|^2 + \frac{4}{\alpha}\|\overline{\boldsymbol{w}}^{(k)} - \boldsymbol{w}^{(k)}\|^2\right\},$$

where the last inequality uses $\alpha, \beta \leq 1$. This establishes the desired lemma. $\qquad\square$

## D    Proof of Theorem 1

**Theorem.** *Assume Assumption 1–2. There exists step size parameters – of the order $\beta = \Theta(1/m), \alpha = \Theta(1/m)$ – such that it holds for any $K \in \mathbb{N}$ that*

$$\mathbb{E}\big[\mathcal{G}(\boldsymbol{\theta}^{(\tilde{K})}, \boldsymbol{w}^{(\tilde{K})})\big] \leq \frac{F^{(K)} + \frac{4}{\mu}\Big(3 + 2m\big(2L_{\boldsymbol{w}}^2\alpha + L_{\boldsymbol{\theta}}^2\beta\big)\Big)\mathbb{E}[\|\nabla_{\boldsymbol{w}}\mathcal{L}(\boldsymbol{\theta}^{(0)}, \boldsymbol{w}^{(0)})\|^2]}{K\min\{\alpha, \frac{\beta}{4}\}}, \tag{58}$$

*where $F^{(K)} := \mathbb{E}[\mathcal{L}(\boldsymbol{\theta}^{(0)}, \boldsymbol{w}^{(0)}) - \mathcal{L}(\boldsymbol{\theta}^{(K)}, \boldsymbol{w}^{(K)})]$ and we recall that $\tilde{K}$ is a uniform random variable drawn from $\{1, ..., K\}$.*

*Proof.* Equipped with the lemmas presented in the main text, we proceed by establishing a bound on $\mathbb{E}\big[\mathcal{L}(\boldsymbol{\theta}^{(K)}, \boldsymbol{w}^{(K)}) - \mathcal{L}(\boldsymbol{\theta}^{(0)}, \boldsymbol{w}^{(0)})\big]$ and $\sum_{k=0}^{K}\|\overline{\boldsymbol{w}}^{(k)} - \boldsymbol{w}^{(k)}\|^2$, as follows. Note that for the sake of simplifying notations, we denoted $\overline{\boldsymbol{w}}^{(k)} = \boldsymbol{w}^{(k)} + \alpha\nabla_{\boldsymbol{w}}\mathcal{L}(\boldsymbol{\theta}^{(k)}, \boldsymbol{w}^{(k)})$ as in Appendix B. We shall also drop the expectation operator while keeping in mind that the bounds to be presented hold only in expectation.

We proceed by telescoping with the change in objective value. Combining (27), (28) yields:

$$\mathcal{L}(\boldsymbol{\theta}^{(k+1)}, \boldsymbol{w}^{(k+1)}) - \mathcal{L}(\boldsymbol{\theta}^{(k)}, \boldsymbol{w}^{(k)})$$

$$\leq \left(2\alpha + \frac{\alpha^3 L_{\boldsymbol{w}}^2}{2} - \frac{\mu\alpha^2}{2}\right)\|\nabla_{\boldsymbol{w}}\mathcal{L}(\boldsymbol{\theta}^{(k)}, \boldsymbol{w}^{(k)})\|^2 + \left(L_{\boldsymbol{\theta}} - \frac{1}{2\beta}\right)\|\overline{\boldsymbol{\theta}}^{(k)} - \boldsymbol{\theta}^{(k)}\|^2 \tag{59}$$

$$+ \left(\alpha - \frac{\mu\alpha^2}{2}\right)\|\mathsf{e}_{\boldsymbol{w}}^{(k)}\|^2 + \frac{\beta}{2}\|\mathsf{e}_{\boldsymbol{\theta}}^{(k)}\|^2 + \left(\frac{L_{\boldsymbol{\theta}}}{2} + \alpha L_{\boldsymbol{w}}^2 - \frac{1}{2\beta}\right)\|\boldsymbol{\theta}^{(k+1)} - \boldsymbol{\theta}^{(k)}\|^2$$

Summing up both sides of (59) from $k = 0$ to $k = K - 1$ yields

$$\mathcal{L}(\boldsymbol{\theta}^{(K)}, \boldsymbol{w}^{(K)}) - \mathcal{L}(\boldsymbol{\theta}^{(0)}, \boldsymbol{w}^{(0)})$$

$$\leq \sum_{k=0}^{K-1}\left\{\left(2\alpha + \frac{\alpha^3 L_{\boldsymbol{w}}^2}{2} - \frac{\mu\alpha^2}{2}\right)\|\nabla_{\boldsymbol{w}}\mathcal{L}(\boldsymbol{\theta}^{(k)}, \boldsymbol{w}^{(k)})\|^2 + \left(L_{\boldsymbol{\theta}} - \frac{1}{2\beta}\right)\|\overline{\boldsymbol{\theta}}^{(k)} - \boldsymbol{\theta}^{(k)}\|^2\right. \tag{60}$$

$$\left. + \left(\alpha - \frac{\mu\alpha^2}{2}\right)\|\mathsf{e}_{\boldsymbol{w}}^{(k)}\|^2 + \frac{\beta}{2}\|\mathsf{e}_{\boldsymbol{\theta}}^{(k)}\|^2 + \left(\frac{L_{\boldsymbol{\theta}}}{2} + \alpha L_{\boldsymbol{w}}^2 - \frac{1}{2\beta}\right)\|\boldsymbol{\theta}^{(k+1)} - \boldsymbol{\theta}^{(k)}\|^2\right\}$$

Using Lemma 4 which is a slight modifications of [30, Lemma 2], and under Assumption 3, we obtain

$$\mathbb{E}[\|\mathbf{e}_{\boldsymbol{\theta}}^{(k)}\|^2] \le 2L_{\boldsymbol{\theta}}^2 \mathbb{E}[\Delta^{(k)}], \quad \mathbb{E}[\|\mathbf{e}_{\boldsymbol{w}}^{(k)}\|^2] \le 2L_{\boldsymbol{w}}^2 \mathbb{E}[\Delta^{(k)}]. \tag{61}$$

This simplifies (60) into

$$\begin{aligned}
&\mathcal{L}(\boldsymbol{\theta}^{(K)}, \boldsymbol{w}^{(K)}) - \mathcal{L}(\boldsymbol{\theta}^{(0)}, \boldsymbol{w}^{(0)}) \\
&\le \sum_{k=0}^{K-1} \left\{ \left(2\alpha + \frac{\alpha^3 L_{\boldsymbol{w}}^2}{2} - \frac{\mu\alpha^2}{2}\right) \|\nabla_{\boldsymbol{w}}\mathcal{L}(\boldsymbol{\theta}^{(k)}, \boldsymbol{w}^{(k)})\|^2 + \left(L_{\boldsymbol{\theta}} - \frac{1}{2\beta}\right) \|\overline{\boldsymbol{\theta}}^{(k)} - \boldsymbol{\theta}^{(k)}\|^2 \right. \\
&\quad \left. + \left(L_{\boldsymbol{w}}^2\left(2\alpha - \mu\alpha^2\right) + L_{\boldsymbol{\theta}}^2\beta\right)\Delta^{(k)} + \left(\frac{L_{\boldsymbol{\theta}}}{2} + \alpha L_{\boldsymbol{w}}^2 - \frac{1}{2\beta}\right)\|\boldsymbol{\theta}^{(k+1)} - \boldsymbol{\theta}^{(k)}\|^2 \right\}
\end{aligned} \tag{62}$$

Denote $F^{(K)} := \mathcal{L}(\boldsymbol{\theta}^{(0)}, \boldsymbol{w}^{(0)}) - \mathcal{L}(\boldsymbol{\theta}^{(K)}, \boldsymbol{w}^{(K)})$. Shuffling terms and dropping terms with negative coefficients $-\frac{\mu\alpha^2}{2}$ for $\Delta^{(k)}$ in the above yield

$$\begin{aligned}
&\sum_{k=0}^{K-1} \left\{ \left(\alpha + \frac{\mu\alpha^2}{2} - \frac{\alpha^3 L_{\boldsymbol{w}}^2}{2}\right) \|\nabla_{\boldsymbol{w}}\mathcal{L}(\boldsymbol{\theta}^{(k)}, \boldsymbol{w}^{(k)})\|^2 + \left(\frac{1}{2\beta} - L_{\boldsymbol{\theta}}\right)\|\overline{\boldsymbol{\theta}}^{(k)} - \boldsymbol{\theta}^{(k)}\|^2 \right\} \\
&\le \sum_{k=0}^{K-1} \left\{ 3\alpha\|\nabla_{\boldsymbol{w}}\mathcal{L}(\boldsymbol{\theta}^{(k)}, \boldsymbol{w}^{(k)})\|^2 + \left(\frac{L_{\boldsymbol{\theta}}}{2} + \alpha L_{\boldsymbol{w}}^2 - \frac{1}{2\beta}\right)\|\boldsymbol{\theta}^{(k+1)} - \boldsymbol{\theta}^{(k)}\|^2 \right. \\
&\quad \left. + \left(2L_{\boldsymbol{w}}^2\alpha + L_{\boldsymbol{\theta}}^2\beta\right)\Delta^{(k)} \right\} + F^{(K)}
\end{aligned} \tag{63}$$

Applying Lemma 3 and (33) gives

$$\begin{aligned}
&\sum_{k=0}^{K-1} \left\{ \left(\alpha + \frac{\mu\alpha^2}{2} - \frac{\alpha^3 L_{\boldsymbol{w}}^2}{2}\right) \|\nabla_{\boldsymbol{w}}\mathcal{L}(\boldsymbol{\theta}^{(k)}, \boldsymbol{w}^{(k)})\|^2 + \left(\frac{1}{2\beta} - L_{\boldsymbol{\theta}}\right)\|\overline{\boldsymbol{\theta}}^{(k)} - \boldsymbol{\theta}^{(k)}\|^2 \right\} \\
&\le \alpha\left(3 + 2m\left(2L_{\boldsymbol{w}}^2\alpha + L_{\boldsymbol{\theta}}^2\beta\right)\right) \sum_{k=0}^{K-1} \|\nabla_{\boldsymbol{w}}\mathcal{L}(\boldsymbol{\theta}^{(k)}, \boldsymbol{w}^{(k)})\|^2 \\
&\quad + \left(\frac{L_{\boldsymbol{\theta}}}{2} + \alpha L_{\boldsymbol{w}}^2 + \left(2L_{\boldsymbol{w}}^2\alpha + L_{\boldsymbol{\theta}}^2\beta\right)\frac{m}{\beta} - \frac{1}{2\beta}\right) \sum_{k=0}^{K-1} \|\boldsymbol{\theta}^{(k+1)} - \boldsymbol{\theta}^{(k)}\|^2 + F^{(K)}
\end{aligned} \tag{64}$$

In order to obtain a bound on $\mathcal{G}(\boldsymbol{\theta}^{(k)}, \boldsymbol{w}^{(k)})$ which consists of a weighted sum of $\|\nabla_{\boldsymbol{w}}\mathcal{L}(\boldsymbol{\theta}^{(k)}, \boldsymbol{w}^{(k)})\|^2$ and $\|\boldsymbol{\theta}^{(k+1)} - \boldsymbol{\theta}^{(k)}\|^2$, we upper bound $\sum_{k=0}^{K-1} \|\nabla_{\boldsymbol{w}}\mathcal{L}(\boldsymbol{\theta}^{(k)}, \boldsymbol{w}^{(k)})\|^2$ such that the positive coefficients of $\|\nabla_{\boldsymbol{w}}\mathcal{L}(\boldsymbol{\theta}^{(k)}, \boldsymbol{w}^{(k)})\|^2$ is accounted for. To this end, exploiting Lemma 2 and summing up both sides of (29) from $k = 1$ to $k = K$ gives

$$\begin{aligned}
&(1 - \mu\alpha)\|\nabla_{\boldsymbol{w}}\mathcal{L}(\boldsymbol{\theta}^{(K)}, \boldsymbol{w}^{(K)})\|^2 - \|\nabla_{\boldsymbol{w}}\mathcal{L}(\boldsymbol{\theta}^{(0)}, \boldsymbol{w}^{(0)})\|^2 \\
&\le \sum_{k=0}^{K-1} \left\{ (\alpha^2 - \mu\alpha)\|\nabla_{\boldsymbol{w}}\mathcal{L}(\boldsymbol{\theta}^{(k)}, \boldsymbol{w}^{(k)})\|^2 + \frac{L_{\boldsymbol{w}}^2}{\mu\alpha}\left(\|\boldsymbol{\theta}^{(k+1)} - \boldsymbol{\theta}^{(k)}\|^2 + \alpha^2\|\mathbf{e}_{\boldsymbol{w}}^{(k)}\|^2\right) \right\}
\end{aligned} \tag{65}$$

We set the following parameters

$$\mu/2 \ge \alpha \tag{a1}$$

and letting $Y^{(K)} := \|\nabla_{\boldsymbol{w}}\mathcal{L}(\boldsymbol{\theta}^{(0)}, \boldsymbol{w}^{(0)})\|^2 - (1 - \mu\alpha)\|\nabla_{\boldsymbol{w}}\mathcal{L}(\boldsymbol{\theta}^{(K)}, \boldsymbol{w}^{(K)})\|^2$ to yield

$$\alpha \sum_{k=0}^{K-1} \|\nabla_{\boldsymbol{w}}\mathcal{L}(\boldsymbol{\theta}^{(k)}, \boldsymbol{w}^{(k)})\|^2 \le \frac{2}{\mu}Y^{(K)} + \frac{2L_{\boldsymbol{w}}^2}{\mu^2\alpha} \sum_{k=0}^{K-1} \left(\|\boldsymbol{\theta}^{(k+1)} - \boldsymbol{\theta}^{(k)}\|^2 + \alpha^2\|\mathbf{e}_{\boldsymbol{w}}^{(k)}\|^2\right) \tag{66}$$

Applying Lemma 3 (with (33)) and Lemma 4 gives

$$\left(\alpha - \frac{4L_{\boldsymbol{w}}^4}{\mu^2}m\alpha^2\right) \sum_{k=0}^{K-1} \|\nabla_{\boldsymbol{w}}\mathcal{L}(\boldsymbol{\theta}^{(k)}, \boldsymbol{w}^{(k)})\|^2 \le \frac{2}{\mu}Y^{(K)} + \left(\frac{2L_{\boldsymbol{w}}^2}{\mu^2\alpha} + \frac{2L_{\boldsymbol{w}}^4}{\mu^2}\frac{\alpha m}{\beta}\right) \sum_{k=0}^{K-1} \|\boldsymbol{\theta}^{(k+1)} - \boldsymbol{\theta}^{(k)}\|^2$$

$$\tag{67}$$

Setting
$$\frac{\mu^2}{8L_{\boldsymbol{w}}^2 m} \geq \alpha, \tag{a2}$$
we have
$$\alpha \sum_{k=0}^{K-1} \|\nabla_{\boldsymbol{w}} \mathcal{L}(\boldsymbol{\theta}^{(k)}, \boldsymbol{w}^{(k)})\|^2 \leq \frac{4}{\mu} Y^{(K)} + \Big(\frac{4L_{\boldsymbol{w}}^2}{\mu^2 \alpha} + \frac{4L_{\boldsymbol{w}}^4}{\mu^2} \frac{\alpha m}{\beta}\Big) \sum_{k=0}^{K-1} \|\boldsymbol{\theta}^{(k+1)} - \boldsymbol{\theta}^{(k)}\|^2 \tag{68}$$
Now, suppose the step size satisfies
$$\frac{1}{2\beta} \geq \Big(3 + 2m\big(2L_{\boldsymbol{w}}^2 \alpha + L_{\boldsymbol{\theta}}^2 \beta\big)\Big)\Big(\frac{4L_{\boldsymbol{w}}^2}{\mu^2 \alpha} + \frac{4L_{\boldsymbol{w}}^4}{\mu^2} \frac{\alpha m}{\beta}\Big) + \frac{L_{\boldsymbol{\theta}}}{2} + \alpha L_{\boldsymbol{w}}^2 + \Big(L_{\boldsymbol{w}}^2 \alpha + L_{\boldsymbol{\theta}}^2 \frac{\beta}{2}\Big)\frac{m}{\beta}, \tag{a3}$$
then substituting (68) into (64) yields a negative coefficient in front of the summation $\sum_{k=0}^{K-1} \|\boldsymbol{\theta}^{(k+1)} - \boldsymbol{\theta}^{(k)}\|^2$.

Hence we obtain that:
$$\sum_{k=0}^{K-1} \left\{ \Big(\alpha + \frac{\mu \alpha^2}{2} - \frac{\alpha^3 L_{\boldsymbol{w}}^2}{2}\Big) \|\nabla_{\boldsymbol{w}} \mathcal{L}(\boldsymbol{\theta}^{(k)}, \boldsymbol{w}^{(k)})\|^2 + \Big(\frac{1}{2\beta} - L_{\boldsymbol{\theta}}\Big)\|\overline{\boldsymbol{\theta}}^{(k)} - \boldsymbol{\theta}^{(k)}\|^2 \right\}$$
$$\leq \Big(3 + 2m\big(2L_{\boldsymbol{w}}^2 \alpha + L_{\boldsymbol{\theta}}^2 \beta\big)\Big)\frac{4}{\mu} Y^{(K)} + F^{(K)} \tag{69}$$
Let the step sizes also satisfy
$$\frac{\mu}{2L_{\boldsymbol{w}}^2} \geq \alpha, \quad \frac{1}{4\beta} \geq L_{\boldsymbol{\theta}}. \tag{a4}$$
The above step size choices results in the bound:
$$\min\{\alpha + \frac{\mu \alpha^2}{4}, \frac{\beta}{4}\} \sum_{k=0}^{K-1} \mathcal{G}(\boldsymbol{\theta}^{(k)}, \boldsymbol{w}^{(k)}) \leq \Big(1 + m\big(2L_{\boldsymbol{w}}^2 \alpha + L_{\boldsymbol{\theta}}^2 \beta\big)\Big)\frac{8}{\mu} Y^{(K)} + F^{(K)} \tag{70}$$
Finally, we observe that
$$\mathbb{E}\big[\mathcal{G}(\boldsymbol{\theta}^{(\tilde{K})}, \boldsymbol{w}^{(\tilde{K})})\big] = \frac{\sum_{k=0}^{K-1} \mathcal{G}(\boldsymbol{\theta}^{(k)}, \boldsymbol{w}^{(k)})}{K} \leq \frac{\Big(3 + 2m\big(2L_{\boldsymbol{w}}^2 \alpha + L_{\boldsymbol{\theta}}^2 \beta\big)\Big)\frac{4}{\mu} Y^{(K)} + F^{(K)}}{K \min\{\alpha, \frac{\beta}{4}\}} \tag{71}$$
This proves the sublinear rate of convergence for the nPD-VR algorithm. Lastly, we note that there exists step sizes with $\alpha = \Theta(1/m), \beta = \Theta(1/m)$ such that the required conditions (a0), (a1), (a2), (a3), (a4) hold [see Section D.2]. Therefore, (71) shows the desired sublinear convergence rate of $\mathcal{O}(m/K)$. $\qquad\square$

## D.1 Auxiliary Lemma

**Lemma 4.** *Under Assumption 3. For any $k \geq 0$, it holds*
$$\mathbb{E}[\|e_{\boldsymbol{\theta}}^{(k)}\|^2] \leq 2L_{\boldsymbol{\theta}}^2 \mathbb{E}\big[\Delta^{(k)}\big], \quad \mathbb{E}[\|e_{\boldsymbol{w}}^{(k)}\|^2] \leq 2L_{\boldsymbol{w}}^2 \mathbb{E}\big[\Delta^{(k)}\big]. \tag{72}$$
*where $\Delta^{(k)}$ is defined in (30).*

*Proof.* The proof follows largely from [30, Lemma 4] with the batch size $b = 1$. Denote $\delta_{i_k} = \nabla_{\boldsymbol{\theta}} \mathcal{L}_{i_k}(\boldsymbol{\theta}^{(k)}, \boldsymbol{w}^{(k)}) - \nabla_{\boldsymbol{\theta}} \mathcal{L}_{i_k}(\boldsymbol{\theta}_{i_k}^{(k)}, \boldsymbol{w}_{i_k}^{(k)})$. We observe that $\mathbb{E}[\delta_{i_k}] = \mathbb{E}[\nabla_{\boldsymbol{\theta}} \mathcal{L}(\boldsymbol{\theta}^{(k)}, \boldsymbol{w}^{(k)}) - \mathsf{G}_{\boldsymbol{\theta}}^{(k)}$. Consider the following chain
$$\mathbb{E}[\|e_{\boldsymbol{\theta}}^{(k)}\|^2] = \mathbb{E}[\|\widetilde{\mathsf{G}}_{\boldsymbol{\theta}}^{(k)} - \nabla_{\boldsymbol{\theta}} \mathcal{L}(\boldsymbol{\theta}^{(k)}, \boldsymbol{w}^{(k)})\|^2]$$
$$= \mathbb{E}\big[\|\delta_{i_k} + \mathsf{G}_{\boldsymbol{\theta}}^{(k)} - \nabla_{\boldsymbol{\theta}} \mathcal{L}(\boldsymbol{\theta}^{(k)}, \boldsymbol{w}^{(k)})\|^2\big] = \mathbb{E}\big[\|\delta_{i_k} - \mathbb{E}[\delta_{i_k}]\|^2\big] \leq \mathbb{E}[\|\delta_{i_k}\|^2]. \tag{73}$$
Furthermore, we observe
$$\mathbb{E}[\|\delta_{i_k}\|^2] = \frac{1}{m} \sum_{i=1}^m \mathbb{E}\big[\|\nabla_{\boldsymbol{\theta}} \mathcal{L}_i(\boldsymbol{\theta}^{(k)}, \boldsymbol{w}^{(k)}) - \nabla_{\boldsymbol{\theta}} \mathcal{L}_i(\boldsymbol{\theta}_i^{(k)}, \boldsymbol{w}_i^{(k)})\|^2\big]$$
$$\leq \frac{2L_{\boldsymbol{\theta}}^2}{m} \sum_{i=1}^m \mathbb{E}\big\{\|\boldsymbol{\theta}^{(k)} - \boldsymbol{\theta}_i^{(k)}\|^2 + \|\boldsymbol{w}^{(k)} - \boldsymbol{w}_i^{(k)}\|^2\big\} = 2L_{\boldsymbol{\theta}}^2 \mathbb{E}[\Delta^{(k)}]. \tag{74}$$
We can repeat the same analysis to upper bound $\mathbb{E}[\|e_{\boldsymbol{w}}^{(k)}\|^2]$. $\qquad\square$

## D.2 Explicit Conditions for $\alpha, \beta$

We first define $\overline{L}^2 := 2L_{\boldsymbol{w}}^2 + L_{\boldsymbol{\theta}}^2$, and require that

$$\alpha \geq \beta. \tag{75}$$

Collect the conditions (a0), (a1), (a2), (a4) to obtain the following requirement on $\alpha$,

$$\alpha \leq \min\left\{\frac{\mu}{2}, \frac{\mu^2}{8L_{\boldsymbol{w}}^2 m}, \frac{1}{(16L_{\boldsymbol{w}}^2 + 2)m}\right\}. \tag{b0}$$

Under the above premises, if additionally,

$$\alpha \leq \frac{1}{m} \frac{1}{12\overline{L}^2 + 96L_{\boldsymbol{w}}^2/\mu^2}, \tag{b1}$$

then one has

$$\frac{1}{8} \geq \alpha m \left[\frac{1}{2}\overline{L}^2 + \frac{4L_{\boldsymbol{w}}^2}{\mu^2}(3 + 2m\overline{L}^2\alpha)\right] \tag{76}$$

Subsequently, we analyze (a3) to find the required $\beta$. Again using $\alpha \geq \beta$, the following implies (a3)

$$\frac{1}{2\beta} \geq \left(3 + 2m\overline{L}^2\alpha\right)\left(\frac{4L_{\boldsymbol{w}}^2}{\mu^2\alpha} + \frac{4L_{\boldsymbol{w}}^4}{\mu^2}\frac{\alpha m}{\beta}\right) + \frac{L_{\boldsymbol{\theta}}}{2} + \alpha L_{\boldsymbol{w}}^2 + \frac{\overline{L}^2}{2}\frac{\alpha m}{\beta} \tag{77}$$

which can be implied by

$$\frac{1}{2\beta} \geq \frac{12L_{\boldsymbol{w}}^2}{\mu^2\alpha} + 8m\overline{L}^2\frac{L_{\boldsymbol{w}}^2}{\mu^2} + \frac{L_{\boldsymbol{\theta}}}{2} + \alpha L_{\boldsymbol{w}}^2 + \left(\left(3 + 2m\overline{L}^2\alpha\right)\frac{4L_{\boldsymbol{w}}^4}{\mu^2} + \frac{\overline{L}^2}{2}\right)\frac{\alpha m}{\beta} \tag{78}$$

Suppose that it obeys that

$$\frac{1}{4\beta} \geq \frac{12L_{\boldsymbol{w}}^2}{\mu^2\alpha} \iff \beta \leq \frac{\mu^2}{48L_{\boldsymbol{w}}^2}\alpha, \tag{79}$$

then (a3) is implied by

$$\frac{1}{4\beta} \geq 8m\overline{L}^2\frac{L_{\boldsymbol{w}}^2}{\mu^2} + \frac{L_{\boldsymbol{\theta}}}{2} + \alpha L_{\boldsymbol{w}}^2 + \left(\left(3 + 2m\overline{L}^2\alpha\right)\frac{4L_{\boldsymbol{w}}^4}{\mu^2} + \frac{\overline{L}^2}{2}\right)\frac{\alpha m}{\beta}. \tag{80}$$

Using (76), we have

$$\frac{1}{8\beta} \geq 8m\overline{L}^2\frac{L_{\boldsymbol{w}}^2}{\mu^2} + \frac{L_{\boldsymbol{\theta}}}{2} + \alpha L_{\boldsymbol{w}}^2 \overset{(b0)}{\Longleftarrow} \frac{1}{8\beta} \geq 8m\overline{L}^2\frac{L_{\boldsymbol{w}}^2}{\mu^2} + \frac{L_{\boldsymbol{\theta}}}{2} + \frac{\mu^2}{8m} \tag{81}$$

We thus need

$$\beta \leq \frac{1}{8}\left(8m\overline{L}^2\frac{L_{\boldsymbol{w}}^2}{\mu^2} + \frac{L_{\boldsymbol{\theta}}}{2} + \frac{\mu^2}{8m}\right)^{-1} \tag{82}$$

In summary, the conditions required are

$$\alpha \leq \min\left\{\frac{\mu}{2}, \frac{\mu^2}{8L_{\boldsymbol{w}}^2 m}, \frac{1}{(16L_{\boldsymbol{w}}^2 + 2)m}, \frac{1}{m}\frac{1}{12\overline{L}^2 + 96L_{\boldsymbol{w}}^2/\mu^2}\right\}, \tag{83}$$

$$\beta \leq \min\left\{\alpha, \frac{\mu^2}{48L_{\boldsymbol{w}}^2}\alpha, \frac{1}{4L_{\boldsymbol{\theta}}}, \frac{1}{8}\left(8m\overline{L}^2\frac{L_{\boldsymbol{w}}^2}{\mu^2} + \frac{L_{\boldsymbol{\theta}}}{2} + \frac{\mu^2}{8m}\right)^{-1}\right\}. \tag{84}$$

We observe that it is possible to set $\alpha = \Theta(1/m), \beta = \Theta(1/m)$.