[Reviews · NeurIPS 2019]

Reviewer 1



Quality: The paper studies policy evaluation with smooth nonlinear function approximation in the batch setting. Authors consider the mean square projected bellman error as objective and cast it to saddle-point problem using convex conjugate trick to avoid double sampling (which is used in several works). The obtained saddle point problem is strongly concave with respect to the dual variable and nonconvex with respect to the primal variable. Then, they extend SAGA to devise a variance reduced algorithm and provide it convergence rate to a stationary point. Some detailed comments: - in line 116: the definition if the projection \Pi is wrong. \Pi is rather the orthogonal projection on the tangent space that passes by V_{\theta} and translated to the origin. I encourage the authors to go back the original paper of Maei et al. - in equation (9): the objective J(\theta) should be devided by 1/2 because the conjugate of 1/2||x||^2 is 1/2||x||^2. - Authors seem to use interchangeably stationary point and saddle point. while a stationary point is not necessarily saddle point. Could authors explain why it is the case in their setting? - I encourage authors to show why assumption 1 implies the boundedness of the dual variables iterates. it is not obvious !! - Also I am curious about the convergence rate of the algorithm without variance reduction i.e using vanilla stochastic gradient method. Clarity: The paper is pretty readable. Originality: The paper builds on many previous works: minimizing mean square projected bellman error, convex conjugate trick, SAGA for non convex smooth optimization. But the theoretical result is still novel and interesting. Significance: The paper would be stronger if it includes discussion on the algorithm solution in term of value function error and on practical implementation as the algorithm involves computing a Hessian matrix. Besides, empirical results would be very appreciated.

Reviewer 2



Overall, the paper made significant contribution to both the reinforcement learning community and optimization community. The proposed algorithm is a variant of non-convex SAGA algorithm introduced by [1]. The novelty comes from their proof for the non-convex but strongly concave case. There are several issues which should be addressed: 1, Recasting the policy evaluation as a primal-dual optimization via the Fenchel duality technique is not new. In fact, [2,3,4] have already exploit this reformulation. First, these related work should be referred appropriately. Given these existing work, this contribution in the paper is relatively straightforward. 2, The choice of the \beta plays a vital role to establish the convergence rate. In fact, the finite-step convergence rate highly depends on the choice of \beta. In the proof in Appendix, the beta should satisfy the conditions in Eq. (a0), (a3), and (a4). However, the existence of \beta satisfying such conditions is not explicit checked. Without the existence of such beta, the convergence rate proposed in the paper is not valid. In sum, this is an interesting work to exploit the advanced optimization technique for reinforcement learning problem. However, there are several conditions in the proof should be treated carefully and rigorously. I am happy to raise the score if the authors can address my concerns. 1] S. J. Reddi, S. Sra, B. Poczos, and A. J. Smola. Proximal stochastic methods for nonsmooth nonconvex finite-sum optimization. In Advances in Neural Information Processing Systems, pages 1145–1153, 2016. [2] B. Liu, J. Liu, M. Ghavamzadeh, S. Mahadevan, and M. Petrik. Finite-sample analysis of proximal gradient TD algorithms. In Conference on Uncertainty in Artificial Intelligence, pages 504–513, 2015. [3] Dai, Bo, He, Niao, Pan, Yunpeng, Boots, Byron, and Song, Le. Learning from conditional distributions via dual embeddings. arXiv:1607.04579, 2016. [4] S. S. Du, J. Chen, L. Li, L. Xiao, and D. Zhou. Stochastic variance reduction methods for policy evaluation. In International Conference on Machine Learning, pages 1049–1058, 2017.

Reviewer 3



This paper studied an off-line TD algorithms via min-max optimization and provided a non-asymptotic analysis of the convergence rate. I do not recommend to accept the paper due to the following reasons. - In my opinion, the paper analyzed only a generic finite-sum min-max optimization problem, not a real analysis for TD algorithm. Here is my reasoning. Although the paper motivates the problem by TD learning, its actual formulation of the problem starts by assuming that a trajectory of state-action sequence is given, and is fixed and deterministic throughout the analysis. Then the problem becomes a typical finite-sum min-max optimization. Throughout the analysis, the randomness of the state and action variables due to their generation via a Markov chain does not play a role, because these variables are treated as fixed (the authors can clarify this in the rebuttal process). Hence, the analysis and results do not reflect any special property of TD algorithms but only min-max minimization. - I only view that paper provides the convergence analysis for the finite-sum min-max problem. For this, the paper proposed variance reduced gradient type of algorithm and characterized the convergence rate. While I do think this makes a contribution, the proof mainly follows from the standard techniques. And such a contribution is not sufficient for publication at NeurIPS. --------------------- After authors' feedback The authors' response did not provide a good answer to my question about their contributions on TD learning. The paper solves only a finite-sum min-max optimization problem, but the claim of contributions (in the title, abstract, Section 1 of introduction, Section 2 of problem setup, etc) was on policy evaluation via nonlinear function approximation in reinforcement learning, which is a very different and much more challenging problem. This is quite misleading for readers to understand the true contribution here. The way that this paper addresses the TD learning problem (see more detail below) does not fit it into the state of the art of TD learning analysis. For the possible interest of the authors, I explain below why I think that the paper does not address the TD learning problem. TD learning has the goal of learning a value function for a policy, and the paper starts from a valid population (in expectation) objective function eq (8) and eq (10) to achieve such a goal. It has been shown (by existing literature) that the solution of such an objective function (eq (8) or eq (10)) does provide desired value function (a good enough approximation to the true value function in the function space). However, the paper does not solve eq (10), but in fact only solves a finite-sum problem eq (16), which is based on a FIXED state-action sequence. The convergence guarantee is established to show the algorithm converges to the saddle point of the finite-sum objective eq (16). Clearly we wonder whether such a solution is desirable for TD learning, i.e., how well such a solution approximates the true value function. There is no such an answer in the paper. One would naturally think that this can be argued by bridging the solution of eq (16) and solution of the original TD objective eq (10), but the paper does not fill this gap. This can be clearly seen from the fact that the analysis of the paper does not even exploit the statistical distribution of state-action and without incorporating this I don't see a way to connect eq (10) and eq (16). Consequently, the paper does not solve the real challenge in the nonlinear TD learning, where we wonder whether we obtain a good enough approximation to the true value function or just a local optimal solution for the chosen objective function.

[Author Response · NeurIPS 2019]

The authors would like to thank all the three reviewers for their useful feedback and the area chair for handling this
paper. To address the reviewers' comments, upon acceptance of this paper, we will *(i) include numerical experiment*
*results, (ii) provide an explicit bound on* $\alpha$, $\beta$, *and (iii) improve clarity.* Some common concerns are as follows.

•• **Empirical Results**: To illustrate the pratical performance of nPD-
VR, we experimented with MountainCar dataset w/ $m = 5000$. We
ran Sarsa to obtain a good policy, then we generate a trajectory of the
state-action pairs. For the nonlinearity, we parametrize $V$ as a two-layer
neural network with $n$ hidden neurons. We set $\gamma = 0.95$, $\alpha = 10^{-4}$,
$\beta = 10^{-8}$ and constraint as $\boldsymbol{\theta} \in \Theta = [0,1]^n$, $\boldsymbol{w} \in [0,100]^n$. Trajectory
of the objective $\mathcal{L}(\boldsymbol{\theta}^{(k)}, \boldsymbol{w}^{(k)})$ is shown on the right. The objective of
nPD-VR converges to (close to) zero in 4-5 passes on data, while a single

(Left) $n = 50$ neurons (Right) $n = 100$ neurons.

timescale SGD on (16) takes a long time (or fail) to converge. Details of this experiment will be found in final version.

**Reviewer 1**: We thank the reviewer for providing constructive and supportive comments.

**Typos**: We apologize for the typos made. **Firstly**, thanks to your suggestions, we have corrected the definition of the
projection $\Pi$ and fixed the constant for the cost function. **Secondly**, we clarify that our algorithm only guarantees
approximate *stationary points* to the *saddle point* MSPBE problem. They will be corrected in the final version.

**Assumption 1**: We acknowledge that it is not obvious to check. An intuition is that as $\Theta$ is bounded and the objective
is strongly concave in $\boldsymbol{w}$, the dual update (w.r.t. $\boldsymbol{w}$) at the $k$th iteration pulls the dual variable towards $\boldsymbol{w}^\star(\boldsymbol{\theta}^{(k)})$, the
unique maximizer given $\boldsymbol{\theta}^{(k)}$, this suggests $\boldsymbol{w}^{(k)}$ may stay in a bounded set. Details will be provided in the final version.

**Complexity**: We will include a comparison to single-timescale primal-dual SGD in terms of computation and memory.
The per iteration computation complexity for both methods are $\mathcal{O}(d^2)$ (due to Hessian-gradient mult.), and the memory
requirement is $\mathcal{O}(d)$ for SGD and $\mathcal{O}(md)$ for nPD-VR – we only need to store $\boldsymbol{\theta}_i^{(k)}, \boldsymbol{w}_i^{(k)}$ as in (20). Convergence
speed for nPD-VR is $\mathcal{O}(1/K)$ while the SGD is only anticipated to converge at $\mathcal{O}(1/\sqrt{K})$ (no known result in the
literature in this setting). The $\mathcal{O}(d^2)$ complexity may appear impractical, yet we can apply a diagonal approximation.

**Reviewer 2**: We thank the reviewer for providing constructive and supportive comments.

**Contributions w.r.t. Related Work**: The suggested references are useful and will be included. We remark that
[1]-[4] only considered linear function approximation, while this work focuses on the nonlinear setting. Also, the fast
convergence of primal-dual SAGA for *one-sided non-convex* problem is new even to the optimization community.

**Existence of** $\beta$: We checked (a0),(a1),(a2),(a3),(a4) carefully and derived this:

$$\alpha \le \min\left\{\frac{\mu^2}{8L_{\boldsymbol{w}}^2 m}, \frac{1/m}{(16L_{\boldsymbol{w}}^2 + 2)}, \frac{1/m}{12\overline{L}^2 + 96L_{\boldsymbol{w}}^2/\mu^2}\right\}, \ \beta \le \min\left\{\frac{\mu^2}{48L_{\boldsymbol{w}}^2}\alpha, \frac{1}{8}\left(8m\overline{L}^2\frac{L_{\boldsymbol{w}}^2}{\mu^2} + \frac{L_\theta}{2} + \frac{\mu^2}{8m}\right)^{-1}\right\}.$$

$\overline{L}^2 = 2L_{\boldsymbol{w}}^2 + L_\theta^2$ and loose constraints are skipped. To get $\alpha, \beta$, we first fix $\alpha$ with the first inequality, then obtain $\beta$.

**Reviewer 3**: We emphasize that our contributions are substantial, from both TD learning and optimization perspectives:

First, we disagree that our paper does not provide a *real analysis for TD algorithm*. Our (10) is actually a TD learning
problem, and we focus on tackling its **batch/offline version (16)**. While in this setting the randomness in state/action
becomes decoupled from the learning process, there are **vast applications** related scenarios with offline available data
and experience replay – as studied in "Batch Reinforcement Learning" by Lange et al., "Least squares policy evaluation
algorithms with linear function approximation" by Nedić et al., etc. (references will be added). All these works only
focused on *linear function approximation*, while we study the **nonlinear** case. Nonlinear TD learning is a challenging
problem due to **non-convexity** in the underlying optimization. It has only been studied by a few authors, e.g., [4,7],
and there are **no prior finite-time analysis** papers. While we consider the batch/offline setting (16), we developed
an algorithm with finite-time analysis and is **efficient**. This result is one of the first in the literature and advances the
analytical understanding for TD learning.

We also disagree that our analysis follows from *standard techniques* in optimization. While the use of variance reduction
(VR) on finite-sum problems is common, applying and analyzing VR on **one-sided non-convex primal dual problem**
(that arises from nonlinear TD) is **new** and **non-trivial**, even to the optimization community. The only comparable
results are the recent works [22,24,27] with focuses on two-timescale and batch update methods. Our novelty is also
evidenced in the analysis in Appendix D where we developed novel analysis techniques to handle the unique challenges.

An **online nonlinear TD learning** algorithm, that accounts for Markovian randomness in state/action, is an interesting
extension. It relates to a stochastic approximation scheme [arXiv:1806.02450] for bilevel programs [arXiv:1802.02246],
and a finite time analysis is possible. This, however, belongs to a different setting than our focus.

[Meta-Review · NeurIPS 2019]

The main contribution of this paper is in solving the finite-sum minimax problem arising from off-line policy evaluation with nonlinear function approximation. The minimax problem is non-convex in the primal variable and strong convexity in the dual subproblem, and a single time-scale algorithm is proposed to find an approximate stationary point. Although it does not address the full stochastic TD learning problem, the progress in the finite-sum off-line version is quite meaningful.